# Exploring Example Influence in Continual Learning

**Qing Sun**[*], **Fan Lyu**[*], **Fanhua Shang**, **Wei Feng**, **Liang Wan**[†]
College of Intelligence and Computing, Tianjin University
{sssunqing, fanlyu, fhshang, wfeng, lwan}@tju.edu.cn
https://github.com/SSSunQing/Example_Influence_CL

## Abstract

Continual Learning (CL) sequentially learns new tasks like human beings, with the goal to achieve better Stability (S, remembering past tasks) and Plasticity (P, adapting to new tasks). Due to the fact that past training data is not available, it is valuable to explore the influence difference on S and P among training examples, which may improve the learning pattern towards better SP. Inspired by Influence Function (IF), we first study example influence via adding perturbation to example weight and computing the influence derivation. To avoid the storage and calculation burden of Hessian inverse in neural networks, we propose a simple yet effective MetaSP algorithm to simulate the two key steps in the computation of IF and obtain the S- and P-aware example influence. Moreover, we propose to fuse two kinds of example influence by solving a dual-objective optimization problem, and obtain a fused influence towards SP Pareto optimality. The fused influence can be used to control the update of model and optimize the storage of rehearsal. Empirical results show that our algorithm significantly outperforms state-of-the-art methods on both task- and class-incremental benchmark CL datasets.

## 1 Introduction

By mimicking human-like learning, Continual Learning (CL) aims to enable a model to continuously learn from novel knowledge (new tasks, new classes, etc.) in a sequential order. The major challenge in CL is to harness catastrophic forgetting and knowledge transition, namely the Stability-Plasticity dilemma, with Stability (S) showing the ability to prevent performance drops for old tasks and Plasticity (P) referring if the new task can be learned rapidly and unimpededly. Intuitively speaking, a robust CL system should achieve outstanding S and P through sequential learning.

The sequential paradigm means CL does not access past training data. Comparing to traditional machine learning, the training data in CL is thus more precious. It is valuable to explore the influence difference on S and P among training examples. Following the accredited influence chain "`Data-Model-Performance`", exploring this difference is equivalent to tracing from performance back to example difference. With appropriate control, this may improve the learning pattern towards better SP. On top of this, *the goal of this paper is to explore the reasonable influence from each training example to SP, and apply the example influence to CL training*.

To understand example influence, one classic successful technique is the Influence Function (IF) [20], which leverages the derivation chain rule from a test objective to training examples. However, directly applying the chain rule leads to computing the inverse of Hessian with the complexity of $O(nq^2 + q^3)$ ($n$ is the number of examples and $q$ is parameter size), which is computationally intensive and may run out-of-memory in neural networks. In this paper, we propose a novel meta-learning algorithm, called MetaSP, to compute example influence via simulating IF. We design based on the rehearsal-based

---

[*]Co-first authors.
[†]Corresponding author.

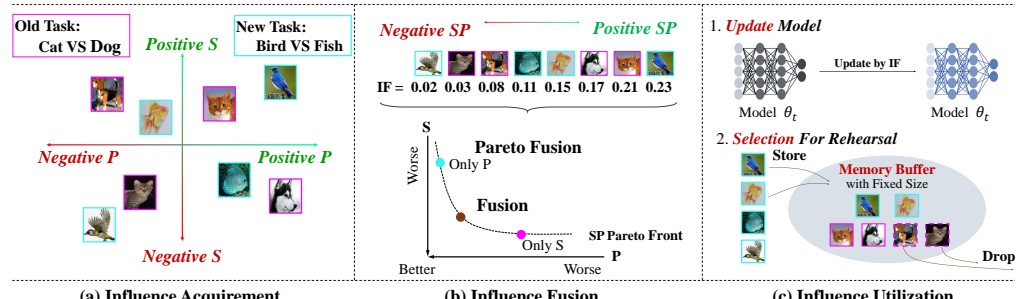

| (a) Influence Acquirement | (b) Influence Fusion | (c) Influence Utilization |

Figure 1: Training examples have different influences on Stability and Plasticity. Given an old task with classes "cat" and "dog" and a new task with classes "Bird" and "Fish", we compute the influence on S and P for each example. Then, we fuse the two kinds of influence towards SP Pareto front. We also show that example influence can be used to adjust model update and optimize rehearsal selection.

CL framework, which avoids forgetting via retraining a part of old data. First, a pseudo update is held with example-level perturbations. Then, two validation sets sampled from seen data are used to compute the gradients on example perturbations. The gradients are regarded as the example influence on S and P. As shown in Fig. 1(a), examples can be distinguished by the value of influence on S and P.

To leverage the two independent kinds of influence in CL, we need to take full account of the influence on both S and P. However, the influence on S and P may interfere with each other, which leads us to make a trade-off. This can be seen as a Dual-Objective Optimization (DOO) problem, which aims to find solutions not dominated (no other better solution) by any other one, i.e. Pareto optimal solutions [8]. We say the solutions as the example influence on SP. Following the gradient-based MGDA algorithm [12], we obtain the fused example influence on SP by meeting the Karush-Kuhn-Tucker (KKT) condition, as illustrated in Figure 1(b).

Finally, we show that the fused influence can be used to control the update of model and optimize the storage of rehearsal in Figure 1(c). On one hand, the fused influence can be directly used to control the magnitude of training loss for each example. On the other hand, under a fixed memory budget, the fused influence can be used to select appropriate examples storing and dropping, which keeps the rehearsal memory always have larger positive influence on SP.

In summary, our contributions are four-fold: 1) Inspired by the influence function, we study CL from the perspective of example difference and propose MetaSP to compute the example influence on S and P. 2) We propose to trade off S and P influence via solving a DOO problem and fuse them towards SP Pareto optimal. 3) We leverage the fused influence to control model update and optimize the storage of rehearsal. 4) The verification contribution: by considering the example influence, in our experiments on both task- and class-incremental CL, better S and more stable P can be observed.

## 2 Related Work

**Continual Learning.** Due to many researchers' efforts, lots of methods for CL have been proposed, which can be classified into three categories. The regularization-based methods [19, 11, 10] are based on regularizing the parameters corresponding to the old tasks and penalizing the feature drift. The parameter isolation based methods [14, 26] generate task-specific parameter expansion or sub-branch. Rehearsal-based methods [29, 7, 22, 6, 1, 2, 28, 3, 25] tackle the challenge of SP dilemma by retaining a subset of old tasks in a stored memory buffer with bounded resources. Although the existing methods apply themselves to achieve better SP, they fail to explore what contributes to the Stability and Plasticity inside the training data. In this work, we explore the problem in the perspective of example difference, where we argue that each example contributes differently to the SP. We focus our work on the rehearsal-based CL framework in order to omit the divergence between models, while evaluating the old data's influence simultaneously.

**Example Influence.** In recent years, as the impressive Interpretable Machine Learning (IML) [27] develops, people realize the importance of exploring the nature of data-driven machine learning. Examples are different, even they belong to the same distribution. Because of such difference, the example contributes differently to the learning pattern. In other words, the influence acquired in advance from different training examples can significantly improve the CL training. Some studies

propose a similar idea and use the influences to reweight or dropout the training data [31, 13, 36]. In contrast to complicated model design, a model-agnostic algorithm estimates the training example influence via computing the derivation from a test loss to a training data weight. One typical example method is the Influence Function [20], which leverages a pure second-order derivation (Hessian) with the chain rule. In this paper, to avoid the expensive computation of Hessian inverse, we design a meta learning [18] based method, which can be used to control the training.

# 3 Demystifying Example Influence on SP

## 3.1 Preliminary: Rehearsal-based CL

Given $T$ different tasks w.r.t. datasets $\{\mathcal{D}_1, \cdots, \mathcal{D}_T\}$, Continual Learning (CL) seeks to learn them in sequence. For the $t$-th dataset (task), $\mathcal{D}_t = \{(x_t^{(n)}, y_t^{(n)})\}_{n=1}^{N_t}$ is split into a training set $\mathcal{D}_t^{\text{trn}}$ and a test set $\mathcal{D}_t^{\text{tst}}$, where $N_t$ is the number of examples. At any time, CL aims at learning a multi-task/multi-class predictor to predict tasks/classes that have been learned (say task-incremental and class-incremental CL). To suppress the catastrophic forgetting, the rehearsal-based CL [30, 22, 32, 6, 16] builds a small size memory buffer $\mathcal{M}_t$ sampled from $\mathcal{D}_t^{\text{trn}}$ for each task (*i.e.*, $|\mathcal{M}_t| \ll |\mathcal{D}_t^{\text{trn}}|$). At training phase, the data in the whole memory $\mathcal{M} = \cup_{k<t}\mathcal{M}_k$ will be retrained together with the current tasks. Accordingly, a mini-batch training step of task $t$ in rehearsal-based CL is denoted as

$$\min_{\boldsymbol{\theta}_t} \quad \ell(\mathcal{B}_{\text{old}} \cup \mathcal{B}_{\text{new}}, \boldsymbol{\theta}_t), \quad \mathcal{B}_{\text{old}} \subset \mathcal{M} \text{ and } \mathcal{B}_{\text{new}} \subset \mathcal{D}_t^{\text{trn}}, \tag{1}$$

where $\ell$ is the empirical loss. $\boldsymbol{\theta}_t$ is the trainable parameters at task $t$ and is updated from scratch.

## 3.2 Example Influence on Stability and Plasticity

**Definition 1 (Stability and Plasticity)** *Suppose the parameter of a model is initialized to $\boldsymbol{\theta}_0$. At the training on the $t$-th task, given test sets of an old task $\mathcal{D}_k^{\text{tst}}(k < t)$ and the current task $\mathcal{D}_t^{\text{tst}}$, the Stability $S_t^k$ and Plasticity $P_t$ can be evaluated by:*

$$S_t^k = p(\mathcal{D}_k^{\text{tst}}|\boldsymbol{\theta}_{t-1}, \mathcal{D}_t^{\text{trn}}) - p(\mathcal{D}_k^{\text{tst}}|\boldsymbol{\theta}_k), \quad P_t = p(\mathcal{D}_t^{\text{tst}}|\boldsymbol{\theta}_{t-1}, \mathcal{D}_t^{\text{trn}}) - p(\mathcal{D}_t^{\text{tst}}|\boldsymbol{\theta}_{t-1}),$$

*where $p(\mathcal{D}_1|\boldsymbol{\theta}, \mathcal{D}_2)$ represents the performance (accuracy in classification) of $\mathcal{D}_1$ conditioned to the model $\boldsymbol{\theta}$ training on $\mathcal{D}_2$. $p(\mathcal{D}|\boldsymbol{\theta})$ denotes the performance of $\mathcal{D}$ tested on the model $\boldsymbol{\theta}$.*

The S of a task is evaluated by the performance difference on the test set after training on any later tasks, which is also known as Forgetting [6]. The P of a task is defined as the ability to integrate new knowledge, which is regarded as the test performance of this task. As many existing CL methods demonstrate, the SP inevitably interferes mutually.

**Definition 2 (Example Influence on SP)** *At the training on the $t$-th task, with a sampled example $x^{\text{trn}} \in \mathcal{D}_t^{\text{trn}}$, the example influence from $x^{\text{trn}}$ to Stability $S_t^k$ and Plasticity $P_t$ for $k < t$ can be evaluated by the gap from deleting it then retraining the model:*

$$I_S(\mathcal{D}_k^{\text{tst}}, x^{\text{trn}}) = p(\mathcal{D}_k^{\text{tst}}|\boldsymbol{\theta}_{t-1}, \mathcal{D}_t^{\text{trn}}) - p(\mathcal{D}_k^{\text{tst}}|\boldsymbol{\theta}_{t-1}, \mathcal{D}_t^{\text{trn}}/x^{\text{trn}}),$$
$$I_P(\mathcal{D}_t^{\text{tst}}, x^{\text{trn}}) = p(\mathcal{D}_t^{\text{tst}}|\boldsymbol{\theta}_{t-1}, \mathcal{D}_t^{\text{trn}}) - p(\mathcal{D}_t^{\text{tst}}|\boldsymbol{\theta}_{t-1}, \mathcal{D}_t^{\text{trn}}/x^{\text{trn}}),$$

*where $\mathcal{D}_t^{\text{trn}}/x^{\text{trn}}$ denotes the dataset $\mathcal{D}_t^{\text{trn}}$ without the training example $x^{\text{trn}}$.*

However, deleting every example to compute full influences is impractical due to the highly computational cost. Instead, the performance change can be indicated by the loss change, which leads to a derivable way to approximate the influence:

$$I_S(\mathcal{D}_k^{\text{tst}}, x^{\text{trn}}) \overset{\text{def}}{=} \frac{\partial \ell(\mathcal{D}_k^{\text{tst}})}{\partial \epsilon}, \quad I_P(\mathcal{D}_t^{\text{tst}}, x^{\text{trn}}) \overset{\text{def}}{=} \frac{\partial \ell(\mathcal{D}_t^{\text{tst}})}{\partial \epsilon}, \tag{2}$$

where $\epsilon$ is the weight perturbation to the training example and $\overset{\text{def}}{=}$ means define. This influence can be computed by the Influence Function [20] that will be introduced in the next section.

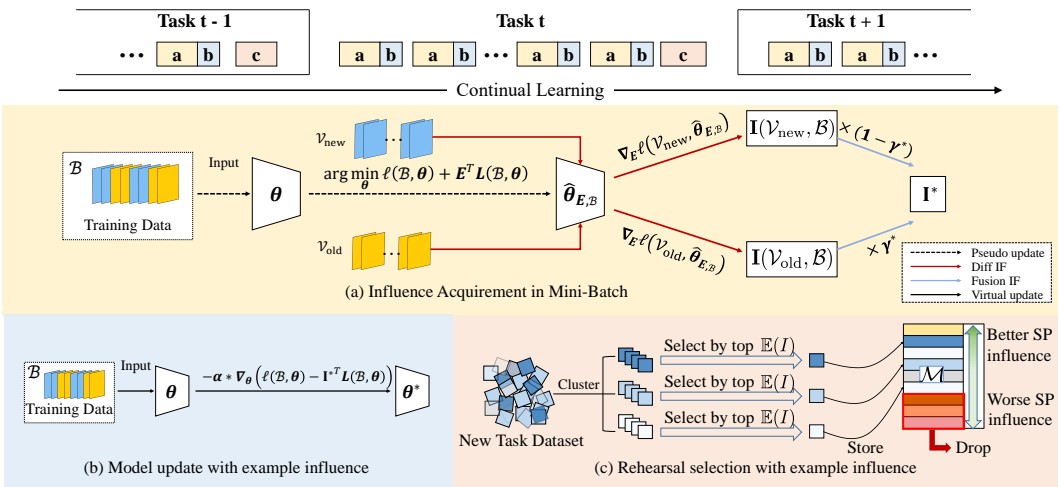

Figure 2: Evaluating and making use of example influence in mini-batch Continual Learning. (a) At each iteration in CL training, MetaSP updates in pseudo and use two validation sets representing old tasks and new task to obtain the example influence on S and P. The two kinds of influence are fused towards a Pareto optimal. (b) The computed influence can be directly used to update CL model and (c) select examples for rehearsal storing and dropping.

## 4 Meta Learning on Stability and Plasticity

### 4.1 Influence Function for SP

A mini-batch, $\mathcal{B}$, from the training data is sampled, and the normal model update is

$$\hat{\boldsymbol{\theta}} = \arg\min_{\boldsymbol{\theta}} \ell\left(\mathcal{B}, \boldsymbol{\theta}\right). \tag{3}$$

In Influence Function (IF) [20], a small weight perturbation $\epsilon$ is added to the training example $x^{\text{trn}} \in \mathcal{B}$

$$\hat{\boldsymbol{\theta}}_{\epsilon,x} = \arg\min_{\boldsymbol{\theta}} \ell\left(\mathcal{B}, \boldsymbol{\theta}\right) + \epsilon\ell(x^{\text{trn}}, \boldsymbol{\theta}), \quad x^{\text{trn}} \in \mathcal{B}. \tag{4}$$

We can easily promote this to the mini-batch

$$\hat{\boldsymbol{\theta}}_{\mathbf{E},\mathcal{B}} = \arg\min_{\boldsymbol{\theta}} \ell\left(\mathcal{B}, \boldsymbol{\theta}\right) + \mathbf{E}^{\top}\mathbf{L}(\mathcal{B}, \boldsymbol{\theta}), \tag{5}$$

where $\mathbf{L}$ denotes the loss vector for a mini-batch and $\mathbf{E} \in \mathbb{R}^{|\mathcal{B}| \times 1}$ denotes the perturbation on each example in it. It is easy to know that the example influence $I(\mathcal{D}^{\text{tst}}, \mathcal{B})$ is reflected in the derivative $\nabla_{\mathbf{E}} \ell(\mathcal{D}^{\text{tst}}, \hat{\boldsymbol{\theta}}_{\mathbf{E},x})\big|_{\mathbf{E}=\mathbf{0}}$. By the chain rule, the example influence in IF can be computed by

$$\mathbf{I}(\mathcal{D}^{\text{tst}}, \mathcal{B}) \stackrel{\text{def}}{=} \nabla_{\mathbf{E}} \ell(\mathcal{D}^{\text{tst}}, \hat{\boldsymbol{\theta}}_{\mathbf{E},x})\big|_{\mathbf{E}=\mathbf{0}} = -\nabla_{\boldsymbol{\theta}} \ell(\mathcal{D}^{\text{tst}}, \hat{\boldsymbol{\theta}}) \mathbf{H}^{-1} \nabla_{\boldsymbol{\theta}}^{\top} \mathbf{L}(\mathcal{B}, \hat{\boldsymbol{\theta}}), \tag{6}$$

where $\mathbf{H} = \nabla_{\boldsymbol{\theta}}^2 \ell(\mathcal{B}, \hat{\boldsymbol{\theta}})$ is a Hessian. Unfortunately, the inverse of Hessian requires the complexity $O(|\mathcal{B}|q^2 + q^3)$ and huge storage for neural networks (maybe out-of-memory), which is challenging for efficient training.

In Eq. (6), we have $\mathbf{I}(\mathcal{D}^{\text{tst}}, \mathcal{B}) = [I(\mathcal{D}^{\text{tst}}, x^{\text{trn}})|x^{\text{trn}} \in \mathcal{B}]$ and find the loss will get larger if $I(\mathcal{D}^{\text{tst}}, x^{\text{trn}}) > 0$, which means the negative influence on the test set $\mathcal{D}^{\text{tst}}$. Similarly, $I(\mathcal{D}^{\text{tst}}, x^{\text{trn}}) < 0$ means the positive influence on the test set $\mathcal{D}^{\text{tst}}$. *Fortunately, the second-order derivation in IF is not necessary under the popular meta learning paradigm such as [18], instead we can easily get the derivation like IF through a one-step pseudo update.* In the following, we will introduce a simple yet effective meta-based method, named MetaSP, to simulate IF at each step with a two-level optimization to avoid computing Hessian inverse.

### 4.2 Simulating IF for SP

Based on the meta learning paradigm, we transform the example influence computation into solving a meta gradient descent problem, named **MetaSP**. For each training step in a rehearsal-based CL,

**Algorithm 1:** Computation of Example Influence (**MetaSP**)

---
**Input:** $\mathcal{B}_{\text{old}}, \mathcal{B}_{\text{new}}, \mathcal{V}_{\text{old}}, \mathcal{V}_{\text{new}}$ ;           `// Training batches, Validation batches`
**Output:** $\mathbf{I}^*$ ;                     `// Pareto example influence on SP`

1 $\hat{\boldsymbol{\theta}}_{\mathbf{E},\mathcal{B}} = \arg\min_{\boldsymbol{\theta}} \; \ell(\mathcal{B}_{\text{old}} \cup \mathcal{B}_{\text{new}}, \boldsymbol{\theta}) + \mathbf{E}^\top \mathbf{L}(\mathcal{B}_{\text{old}} \cup \mathcal{B}_{\text{new}}, \boldsymbol{\theta})$ ;     `// Pseudo update`
2 $\mathbf{I}(\mathcal{V}_{\text{old}}, \mathcal{B}) = \nabla_{\mathbf{E}} \ell(\mathcal{V}_{\text{old}}, \hat{\boldsymbol{\theta}}_{\mathbf{E},\mathcal{B}})$ ;          `// Gradient from old val loss`
3 $\mathbf{I}(\mathcal{V}_{\text{new}}, \mathcal{B}) = \nabla_{\mathbf{E}} \ell(\mathcal{V}_{\text{new}}, \hat{\boldsymbol{\theta}}_{\mathbf{E},\mathcal{B}})$;          `// Gradient from new val loss`
4 $\gamma^* \leftarrow$ Eq. (11);                `// Optimal fusion hyper-parameter`
5 $\mathbf{I}^* = \gamma^* \cdot \mathbf{I}(\mathcal{V}_{\text{old}}, \mathcal{B}) + (1 - \gamma^*) \cdot \mathbf{I}(\mathcal{V}_{\text{new}}, \mathcal{B})$;         `// Influence fusion`

---

we have two mini-batches data $\mathcal{B}_{\text{old}}$ and $\mathcal{B}_{\text{new}}$ in respect to old and new tasks. Our goal is to obtain the influence on S and P from every example in $\mathcal{B}_{\text{old}} \cup \mathcal{B}_{\text{new}}$. Note that both S-aware and P-aware influence are applied to every example regardless of old or new tasks. That is, the contribution of an example is not deterministic. Data of old tasks may also affect the new task in positive, and vice-versa. In rehearsal-based CL, we turn to computing the derivations $\nabla_{\mathbf{E}} \ell(\mathcal{V}_{\text{old}}, \hat{\boldsymbol{\theta}})|_{\mathbf{E}=0}$ for example influence.

To compute the derivation, as shown in Fig. 2(a), our MetaSP has two key steps:

**(1) Pseudo update**. This step is to simulate Eq. (5) in IF via a pseudo update

$$\hat{\boldsymbol{\theta}}_{\mathbf{E},\mathcal{B}} = \arg\min_{\boldsymbol{\theta}} \quad \ell(\mathcal{B}_{\text{old}} \cup \mathcal{B}_{\text{new}}, \boldsymbol{\theta}) + \mathbf{E}^\top \mathbf{L}(\mathcal{B}_{\text{old}} \cup \mathcal{B}_{\text{new}}, \boldsymbol{\theta}), \qquad (7)$$

where $\mathbf{L}$ denotes the loss vector for a mini-batch combining both old and new tasks.

**(2) Compute example influence**. This step computes example influence on S and P for all training examples as simulating Eq. (6). Based on the pseudo updated model in Eq. (7), we compute S- and P-aware example influence via two validation sets $\mathcal{V}_{\text{old}}$ and $\mathcal{V}_{\text{new}}$. Noteworthily, because the test set $\mathcal{D}^{\text{tst}}$ is unavailable at training phase, we use two dynamic validation sets $\mathcal{V}_{\text{old}}$ and $\mathcal{V}_{\text{new}}$ to act as the alternative in the CL training process. One is sampled from the memory buffer ($\mathcal{V}_{\text{old}}$) representing the old tasks, and the other is from the seen training data representing the new task ($\mathcal{V}_{\text{new}}$). With $\mathbf{E}$ initialized to $\mathbf{0}$, the two kinds of example influence are computed as

$$\mathbf{I}(\mathcal{V}_{\text{old}}, \mathcal{B}) = \nabla_{\mathbf{E}} \ell(\mathcal{V}_{\text{old}}, \hat{\boldsymbol{\theta}}_{\mathbf{E},\mathcal{B}}), \quad \mathbf{I}(\mathcal{V}_{\text{new}}, \mathcal{B}) = \nabla_{\mathbf{E}} \ell(\mathcal{V}_{\text{new}}, \hat{\boldsymbol{\theta}}_{\mathbf{E},\mathcal{B}}). \qquad (8)$$

Generally, each elements in two influence vectors $\mathbf{I}(\mathcal{V}_{\text{old}}, \mathcal{B})$ and $\mathbf{I}(\mathcal{V}_{\text{new}}, \mathcal{B})$ represents the example influence on S and P. Similar to IF, elements with positive value mean negative influence while elements with negative value mean positive influence.

## 5 Using Influence for Continual Learning

### 5.1 Before Using: Influence for SP Pareto Optimality

As shown in Eq. (8), the example influence is equal to the derivation from validation loss of old and new tasks to the perturbations $\mathbf{E}$. However, the two kinds of influence are independent and interfere with each other. That is, using only one of them may fail the other performance. We prefer to find a solution that makes a trade-off between the influence on both S and P. Thus, we integrate the two influence $\mathbf{I}(\mathcal{V}_{\text{old}}, \mathcal{B})$ and $\mathbf{I}(\mathcal{V}_{\text{new}}, \mathcal{B})$ into a DOO problem with two gradients from different objectives.

$$\min_{\mathbf{E}} \quad \left\{ \ell(\mathcal{V}_{\text{old}}, \hat{\boldsymbol{\theta}}_{\mathbf{E},\mathcal{B}}), \ell(\mathcal{V}_{\text{new}}, \hat{\boldsymbol{\theta}}_{\mathbf{E},\mathcal{B}}) \right\}. \qquad (9)$$

The goal of Problem (9) is to obtain a fused way that satisfies the SP Pareto optimality.

**Definition 3 (SP Pareto Optimality)**

*1. (**Pareto Dominate**) Let $\mathbf{E}_a$, $\mathbf{E}_b$ be two solutions for Problem (9), $\mathbf{E}_a$ is said to dominate $\mathbf{E}_b$ ($\mathbf{E}_a \prec \mathbf{E}_b$) if and only if $\ell(\mathcal{V}, \hat{\boldsymbol{\theta}}_{\mathbf{E}_a,\mathcal{B}}) \leq \ell(\mathcal{V}, \hat{\boldsymbol{\theta}}_{\mathbf{E}_b,\mathcal{B}})$, $\forall \mathcal{V} \in \{\mathcal{V}_{\text{old}}, \mathcal{V}_{\text{new}}\}$, and $\ell(\mathcal{V}, \hat{\boldsymbol{\theta}}_{\mathbf{E}_a,\mathcal{B}}) < \ell(\mathcal{V}, \hat{\boldsymbol{\theta}}_{\mathbf{E}_b,\mathcal{B}})$, $\exists \mathcal{V} \in \{\mathcal{V}_{\text{old}}, \mathcal{V}_{\text{new}}\}$.*
*2. (**SP Pareto Optimal**) $\mathbf{E}$ is called SP Pareto optimal if no other solution can have better values in $\ell(\mathcal{V}_{old}, \hat{\boldsymbol{\theta}}_{\mathbf{E},\mathcal{B}})$ and $\ell(\mathcal{V}_{new}, \hat{\boldsymbol{\theta}}_{\mathbf{E},\mathcal{B}})$.*

**Algorithm 2:** Using Example Influence in Rehearsal-based Continual Learning.

**Input:** Initialized $\boldsymbol{\theta}_0$, Learning rate $\alpha$, Training set $\{\mathcal{D}_1^{\text{trn}}, \cdots, \mathcal{D}_T^{\text{trn}}\}$, Memory $\mathcal{M}$
**Output:** $\boldsymbol{\theta}_T$ ;                                                          // Final model

1 **for** *task* $t = 1 : T$ **do**
2    $\boldsymbol{\theta}_t = \text{TrainNewTask}(\boldsymbol{\theta}_{t-1}, \mathcal{D}_t^{\text{trn}}, \mathcal{M})$ (Alg. 3)    6    **for** $i = 1 : \frac{|\mathcal{M}|}{t}$ **do**
3    $\mathcal{C}_1, \mathcal{C}_2, \cdots, \mathcal{C}_{\frac{|\mathcal{M}|}{t}} \leftarrow \text{K-Means}(\mathcal{D}_t^{\text{trn}})$;    7    Pop the bottom of $\mathcal{M}$;
4    Rank $\mathcal{C}_i$ with $\mathbb{E}(I^*(x)), x \in \mathcal{C}_i$;    8    Push the top of $\mathcal{C}_i$ to $\mathcal{M}$;
5    Rank $\mathcal{M}$ with $\mathbb{E}(I^*(x)), x \in \mathcal{M}$;    9    **end**
10 **end**

Inspired by the Multiple-Gradient Descent Algorithm (MGDA) [12], we transform Problem (9) to a min-norm problem. Specifically, according to the KKT conditions [15], we have

$$\gamma^* = \arg\min_\gamma \left\| \gamma \nabla_{\mathbf{E}} \ell(\mathcal{V}_{\text{old}}, \hat{\boldsymbol{\theta}}_{\mathbf{E}, \mathcal{B}}) + (1 - \gamma) \nabla_{\mathbf{E}} \ell(\mathcal{V}_{\text{new}}, \hat{\boldsymbol{\theta}}_{\mathbf{E}, \mathcal{B}}) \right\|_2^2, \quad s.t., 0 \le \gamma \le 1. \tag{10}$$

Referring to the study from Sener *et al.* [34], the optimal $\gamma^*$ is easily computed as

$$\gamma^* = \min\left( \max\left( \frac{(\nabla_{\mathbf{E}} \ell(\mathcal{V}_{\text{new}}, \hat{\boldsymbol{\theta}}_{\mathbf{E}, \mathcal{B}}) - \nabla_{\mathbf{E}} \ell(\mathcal{V}_{\text{old}}, \hat{\boldsymbol{\theta}}_{\mathbf{E}, \mathcal{B}}))^\top \nabla_{\mathbf{E}} \ell(\mathcal{V}_{\text{new}}, \hat{\boldsymbol{\theta}}_{\mathbf{E}, \mathcal{B}})}{\|\nabla_{\mathbf{E}} \ell(\mathcal{V}_{\text{new}}, \hat{\boldsymbol{\theta}}_{\mathbf{E}, \mathcal{B}}) - \nabla_{\mathbf{E}} \ell(\mathcal{V}_{\text{old}}, \hat{\boldsymbol{\theta}}_{\mathbf{E}, \mathcal{B}})\|_2^2}, 0 \right), 1 \right). \tag{11}$$

Thus, the SP Pareto influence of the training batch can be computed by

$$\mathbf{I}^* = \gamma^* \cdot \mathbf{I}(\mathcal{V}_{\text{old}}, \mathcal{B}) + (1 - \gamma^*) \cdot \mathbf{I}(\mathcal{V}_{\text{new}}, \mathcal{B}). \tag{12}$$

This process can be seen in Fig. 2(a). Different from the S-aware and P-aware influence, the integrated influence consider the Pareto optimum to both S and P, *i.e.*, reducing the negative influence on S or P and keeping the positive influence on both S and P. Then we will introduce how to leverage example influence in CL training, our algorithm can be seen in Alg. 1.

### 5.2 Model Update Using Example Influence

With the computed example influence in each mini-batch, we can easily control the model update of this mini-batch to adjust the training towards an ensemble positive direction. Given parameter $\boldsymbol{\theta}$ from the previous iteration the step size $\alpha$, the model can be updated in traditional SGD as $\boldsymbol{\theta}^* = \boldsymbol{\theta} - \alpha \cdot \nabla_{\boldsymbol{\theta}} (\ell(\mathcal{B}, \boldsymbol{\theta}))$, where $\mathcal{B} = \mathcal{B}_{\text{old}} \cup \mathcal{B}_{\text{new}}$. By regularizing the update with the example influence $\mathbf{I}^*$, we have

$$\boldsymbol{\theta}^* = \boldsymbol{\theta} - \alpha \cdot \nabla_{\boldsymbol{\theta}} \left( \ell(\mathcal{B}, \boldsymbol{\theta}) + (-\mathbf{I}^*)^\top \mathbf{L}(\mathcal{B}, \boldsymbol{\theta}) \right). \tag{13}$$

MetaSP offers regularized updates at every step for rehearsal-based CL, which leads the CL training to better SP but with only the complexity of $O(|\mathcal{B}|q + vq)$ ($v$ denotes the validation size) compared with that of IF, $O(|\mathcal{B}|q^2 + q^3)$.

**Algorithm 3:** Training New Task

**Input:** Initialized $\boldsymbol{\theta}_t$, Training set $\mathcal{D}_t^{\text{trn}}$,
        Memory $\mathcal{M}$, Learning rate $\alpha$
**Output:** Trained $\boldsymbol{\theta}_t$

1 **for** $i = 1 : ITER\_NUM$ **do**
2    $\mathcal{B}_{\text{new}} \sim \mathcal{D}_t^{\text{trn}}$;
3    **if** $t = 1$ **then**
4      $\boldsymbol{\theta}_t = \boldsymbol{\theta}_t - \alpha \cdot \nabla_{\boldsymbol{\theta}} \ell(\mathcal{B}_{\text{new}}, \boldsymbol{\theta}_t)$;
5    **else**
6      $\mathcal{B}_{\text{old}} \sim \mathcal{M}, \mathcal{V}_{\text{old}} \sim \mathcal{M}, V_{\text{new}} \sim \mathcal{D}_t^{\text{trn}}$;
7      $\mathbf{I}^* \leftarrow \text{METASP}(\mathcal{B}_{\text{old}}, \mathcal{B}_{\text{new}}, \mathcal{V}_{\text{old}}, \mathcal{V}_{\text{new}})$;
8      $\boldsymbol{\theta}_t = \boldsymbol{\theta}_t - \alpha \cdot \nabla_{\boldsymbol{\theta}} (\ell(\mathcal{B}_{\text{old}} \cup \mathcal{B}_{\text{new}}, \boldsymbol{\theta}_t)$
9         $+ (-\mathbf{I}^*)^\top \mathbf{L}(\mathcal{B}_{\text{old}} \cup \mathcal{B}_{\text{new}}, \boldsymbol{\theta}_t))$;
10    **end**
11 **end**

We show this application in Fig. 2(b). By updating like the above equation, we can make use of the influence of each example to a large extent. In this way, some useless examples are restrained and some positive examples are emphasized, which may improve the acquisition of new knowledge and the maintenance of old knowledge simultaneously.

### 5.3 Rehearsal Selection Using Example Influence

Rehearsal in fixed budget needs to consider *storing* and *dropping* to keep the memory $\mathcal{M}$ having the core set of all old tasks. In tradition, storing and dropping are both based on randomly example

selection, which ignores the influence difference on SP from each example. Given influence $I^*(x)$ representing contributions from example $x$ to SP, we further design to use it to improve the rehearsal strategy under fixed memory budget. The above example influence on S and P is computed in mini-batch level, we can promote it to the whole dataset according to the law of large numbers, and the influence value for the example $x$ is the value of expectation over batches, *i.e.*, $\mathbb{E}(I^*(x))$.

The fixed-size memory is divided averagely by the seen task number. After task $t$ finishes its training, we conduct our influence-aware rehearsal selection strategy as shown in Fig. 2(c). For storing, we first cluster all training data into $\frac{|\mathcal{M}|}{t}$ groups using K-means to diversify the store data. Each group is ranked by its SP influence value, and the most positive influence on both SP will be selected to store. For dropping, we rank again on the memory buffer via their influence value, and drop the most negative $\frac{|\mathcal{M}|}{t}$ example. In this way, $\mathcal{M}$ always stores diverse examples with positive SP influence.

## 6 Experiments

### 6.1 Datasets and implementation details

We use three commonly used benchmarks for evaluation: 1) **Split CIFAR-10** [37] consists of 5 tasks, with 2 distinct classes each and 5000 exemplars per class, deriving from the CIFAR-10 dataset; 2) **Split CIFAR-100** [37] splits the original CIFAR-100 dataset into 10 disjoint subsets, each of which is considered as a separate task with 10 classes; 3) **Split Mini-Imagenet** [35] is a subset of 100 classes from ImageNet [9], rescaled to $32 \times 32$. Each class has 600 samples, randomly subdivided into training ($80\%$) and test sets ($20\%$). Mini-Imagenet dataset is equally divided into 5 disjoint tasks.

We employ ResNet-18 [17] as the backbone which is trained *from scratch*. We use Stochastic Gradient Descent (SGD) optimizer and set the batch size 32 unchanged in order to guarantee an equal number of updates. Also, the rehearsal batch sampled from memory buffer is set to 32. We construct the SP validation sets in MetaSP by randomly sampling $10\%$ of the seen data and $10\%$ of the memory buffer at each training step. We set other hyper-settings following ER tricks [4], including 50 total epochs and hyper-parameters. All results are averaged over 5 fixed seeds for fairness.

To better evaluate the CL process, we suggest evaluating SP with four metrics as follows. We use the sign function $\mathbf{1}(\cdot)$ to represent if the prediction of model is equal to the ground truth. 1) **First Accuracy** ($A_1 = \frac{1}{T} \sum_t \sum_{x_i \in \mathcal{D}_t^{\mathrm{tst}}} \mathbf{1}(y_i, \boldsymbol{\theta}_t(x_i))$): For each task, when it is first trained done, we evaluate its testing performance immediately, which indicates the Plasticity, *i.e.*, the capability of learning new knowledge. 2) **Final Accuracy** ($A_\infty = \frac{1}{T} \sum_t \sum_{x_i \in \mathcal{D}_t^{\mathrm{tst}}} \mathbf{1}(y_i, \boldsymbol{\theta}_T(x_i))$): This metric is the final performance for each task, which indicates Stability, *i.e.*, the capability of suppressing catastrophic forgetting. 3) **Mean Average Accuracy** ($A_{\mathrm{m}} = \frac{1}{T} \sum_t \left( \frac{1}{t} \sum_{k \le t} \sum_{x_i \in \mathcal{D}_k^{\mathrm{tst}}} \mathbf{1}(y_i, \boldsymbol{\theta}_t(x_i)) \right)$): This metric computes along CL process, indicating the SP performance after each task trained done. 4) **Backward Transfer** ($BWT = \frac{1}{T-1} \sum_{t=1}^{T-1} \sum_{(x,y) \in \mathcal{D}_t^{\mathrm{tst}}} (\mathbf{1}(y, \theta_T(x)) - \mathbf{1}(y, \theta_t(x))) = \frac{T}{T-1}(A_\infty - A_1)$): This metric is the performance drop from first to final accuracy of each task.

### 6.2 Main Comparison Results

We compare our method against 8 rehearsal-based methods (including GDUMB [28], GEM [22], AGEM [6], HAL [5], GSS [2], MIR [1], GMED [33] and ER [7]). What's more, we also provide a lower bound that train new data directly without any forgetting avoidance strategy (Fine-tune) and an upper bound that is given by all task data through joint training (Joint).

In Table 1, we show the quantitative results of all compared methods and the proposed MetaSP in class-incremental and task-incremental settings. First of all, by controlling the training according to the influence on SP, the proposed MetaSP outperforms other methods on all metrics. With the memory buffer size growth, all the rehearsal-based CL get better performance, while the advantages of MetaSP are more obvious. In terms of the First Accuracy $A_1$, indicating the ability to learn new tasks, our method outperforms most of the other methods with a little numerical advantage. In terms of the Final Accuracy $A_\infty$, which is used to measure the forgetting, we have an obvious improvement of an average of 3.17 for class-incremental setting and averagely 1.77 for task-incremental setting w.r.t. the second best result. This shows MetaSP can significantly keep stable learning of the new task while suppressing the catastrophic forgetting. It is because although the new tasks may have

Table 1: Comparisons on three datasets, averaged across 5 runs (See std. in the Appendix). Red and blue values mean the best in our methods and the compared methods. ● indicates that our method is significantly better than the compared method (paired t-tests at 95% significance level).

| Method | CIFAR10 (Class increment) buffer size 300 | | | | buffer size 500 | | | | CIFAR10 (Task increment) buffer size 300 | | | | buffer size 500 | | | |
|---|---|---|---|---|---|---|---|---|---|---|---|---|---|---|---|---|
| | $A_1$ | $A_\infty$ | $A_m$ | BWT | $A_1$ | $A_\infty$ | $A_m$ | BWT | $A_1$ | $A_\infty$ | $A_m$ | BWT | $A_1$ | $A_\infty$ | $A_m$ | BWT |
| | Finetune | 19.66 | Joint | 91.79 | | | | | Finetune | 65.27 | Joint | 98.16 | | | | |
| GDUMB [28] | 36.92 | | | | 44.27 | | | | 73.22 | | | | 78.06 | | | |
| GEM [22] | 93.90● | 37.51● | 55.43● | −70.48 | 92.76● | 36.95● | 57.36● | −69.76● | 96.62● | 89.34● | 92.49● | −9.09● | 96.73● | 90.42 | 92.93● | −7.88 |
| AGEM [6] | 96.57 | 20.02● | 45.57● | −95.68● | 96.56● | 20.01● | 46.52● | −95.69● | 96.78● | 85.52● | 90.16● | −14.07● | 96.71● | 86.45● | 90.90● | −12.83● |
| HAL [5] | 91.30● | 24.45● | 46.34● | −83.56● | 91.96● | 27.94● | 49.05● | −80.01● | 91.41● | 79.90● | 83.78● | −14.39● | 92.03● | 81.84● | 84.19● | −12.73● |
| MIR [1] | 96.70● | 38.53● | 56.96● | −72.72● | 96.65 | 42.65● | 59.99● | −67.50● | 96.76● | 88.50● | 90.87● | −10.33● | 96.73● | 90.63● | 91.99● | −7.62 |
| GSS [2] | 96.53 | 35.89● | 54.33● | −75.80● | 96.55 | 41.96● | 58.16● | −68.24● | 96.56● | 88.05● | 90.60● | −10.63● | 96.57● | 90.38 | 92.19● | −7.73 |
| GMED [33] | 96.65 | 38.12● | 58.92● | −73.16● | 96.65 | 43.68● | 62.56● | −66.21● | 96.73● | 88.91● | 91.20● | −9.76● | 96.72● | 89.72● | 92.10● | −8.75 |
| ER [7] | 96.73 | 34.19● | 53.72● | −78.18● | 96.74 | 40.45● | 57.69● | −70.36● | 96.93● | 88.97● | 91.12● | −9.95● | 96.79● | 90.60 | 92.28● | −7.74 |
| Ours | 96.87 | 42.42 | 63.52 | −68.05 | 96.82 | 49.16 | 67.88 | −59.57 | 97.10 | 89.40 | 92.54 | −9.62 | 97.31 | 90.91 | 93.38 | −7.99 |
| Ours+RehSel | 96.85 | 43.76 | 63.69 | −66.36 | 96.81 | 50.10 | 68.28 | −58.38 | 97.11 | 89.91 | 92.66 | −8.99 | 97.30 | 91.41 | 93.28 | −7.36 |

| Method | CIFAR100 (Class increment) buffer size 500 | | | | buffer size 1000 | | | | CIFAR100 (Task increment) buffer size 500 | | | | buffer size 1000 | | | |
|---|---|---|---|---|---|---|---|---|---|---|---|---|---|---|---|---|
| | $A_1$ | $A_\infty$ | $A_m$ | BWT | $A_1$ | $A_\infty$ | $A_m$ | BWT | $A_1$ | $A_\infty$ | $A_m$ | BWT | $A_1$ | $A_\infty$ | $A_m$ | BWT |
| | Finetune | 9.14 | Joint | 71.25 | | | | | Finetune | 33.85 | Joint | 91.63 | | | | |
| GDUMB [28] | 11.11 | | | | 15.75 | | | | 36.40 | | | | 43.25 | | | |
| GEM [22] | 85.28● | 15.91● | 29.38● | −77.07 | 84.28● | 22.79● | 34.09● | −68.32 | 85.53● | 68.68● | 68.49● | −18.72 | 85.24● | 73.71● | 72.59● | −12.81 |
| AGEM [6] | 85.97 | 9.31● | 24.60● | −85.18● | 85.66 | 9.27● | 24.67● | −84.88● | 85.97● | 55.28● | 58.23● | −34.10● | 85.66● | 55.95● | 59.96● | −33.01● |
| HAL [5] | 67.33● | 8.20● | 22.72● | −65.70● | 68.06● | 10.59● | 24.74● | −63.86● | 67.64● | 44.98● | 50.79● | −25.17● | 68.62● | 50.07● | 54.01● | −20.61● |
| MIR [1] | 87.38 | 13.49● | 28.88● | −82.09● | 87.39 | 17.56● | 32.48● | −77.59● | 87.42 | 66.18● | 67.43● | −23.60● | 87.50 | 71.20● | 71.42● | −18.10● |
| GSS [2] | 86.03 | 14.01● | 28.00● | −80.02● | 86.31 | 17.87● | 31.82● | −76.04● | 86.10● | 66.80● | 66.55● | −21.44 | 86.44● | 71.98● | 71.00● | −16.06 |
| GMED [33] | 87.18 | 14.56● | 33.41● | −80.68● | 87.29 | 18.67● | 38.69● | −76.23● | 87.30● | 68.82● | 72.66● | −20.53 | 87.49 | 73.91● | 76.36● | −15.10 |
| ER [7] | 87.23 | 13.75● | 28.88● | −81.64● | 87.33 | 17.56● | 32.45● | −77.52● | 87.29 | 66.82● | 67.56● | −22.73 | 87.40 | 71.74● | 71.60● | −17.40● |
| Ours | 88.13 | 18.96 | 38.62 | −76.85 | 87.58 | 24.78 | 45.20 | −69.76 | 88.94 | 70.03 | 74.07 | −21.01 | 88.94 | 75.32 | 78.09 | −15.14 |
| Ours+RehSel | 87.81 | 19.28 | 39.23 | −76.13 | 87.55 | 25.72 | 45.48 | −68.69 | 88.58 | 70.81 | 74.24 | −19.74 | 89.03 | 76.14 | 78.27 | −14.32 |

| Method | Mini-Imagenet (Class increment) buffer size 500 | | | | buffer size 1000 | | | | Mini-Imagenet (Task increment) buffer size 500 | | | | buffer size 1000 | | | |
|---|---|---|---|---|---|---|---|---|---|---|---|---|---|---|---|---|
| | $A_1$ | $A_\infty$ | $A_m$ | BWT | $A_1$ | $A_\infty$ | $A_m$ | BWT | $A_1$ | $A_\infty$ | $A_m$ | BWT | $A_1$ | $A_\infty$ | $A_m$ | BWT |
| | Finetune | 11.12 | Joint | 44.39 | | | | | Finetune | 23.46 | Joint | 62.30 | | | | |
| GDUMB [28] | 6.22 | | | | 7.15 | | | | 16.37 | | | | 17.69 | | | |
| AGEM [6] | 50.06● | 10.69● | 22.29● | −49.22 | 50.03● | 10.69● | 22.28● | −49.16● | 50.06● | 18.34● | 28.05● | −39.65● | 50.03● | 18.78● | 28.12● | −39.05● |
| MIR [1] | 51.44 | 11.07● | 23.65● | −50.46● | 51.25● | 11.32● | 24.09● | −49.92● | 51.47● | 29.10● | 35.20● | −27.95● | 51.31● | 31.39● | 37.24● | −24.89● |
| GSS [2] | 51.63 | 11.09● | 23.62● | −50.66● | 51.35● | 11.42● | 24.05● | −49.91● | 51.64● | 28.67● | 35.22● | −28.71● | 51.40● | 31.75● | 37.23● | −24.56● |
| GMED [33] | 51.21 | 11.03● | 24.47● | −50.23● | 50.87 | 11.73● | 25.50● | −48.93● | 51.29● | 30.47● | 37.64● | −26.02● | 51.00● | 32.85● | 39.66● | −22.69● |
| ER [7] | 51.68 | 11.00● | 23.71● | −50.84● | 51.41 | 11.35● | 24.08● | −50.08● | 51.70 | 28.97● | 35.30● | −28.40● | 51.55● | 31.59● | 37.36● | −24.95● |
| Ours | 51.76 | 12.48 | 26.50 | −49.10 | 50.91 | 14.43 | 28.47 | −45.59 | 52.44 | 32.59 | 39.38 | −24.82 | 52.27 | 36.25 | 41.59 | −20.02 |
| Ours+RehSel | 51.81 | 12.74 | 26.43 | −48.84 | 50.96 | 14.54 | 28.44 | −45.52 | 51.73 | 34.36 | 40.48 | −21.70 | 51.47 | 37.20 | 42.19 | −17.83 |

Figure 3: Top: Statistics of examples with positive and negative influence on S, P, and SP. Bottom: We divide all example influences equally into 5 groups, and count the number in each range.

(a) Total 500 memory data   (b) Total 10000 new data per task

larger gradient to dominant the update for all rehearsal-based CL, our method improves the example with positive effective and restrain the negative-impact example. In terms of the Mean Average Accuracy $A_m$, which evaluates the SP throughout the whole CL process, our method shows its significant superiority with an average improvement of over 4.44 and 1.24 w.r.t the second best results in class-incremental and task-incremental settings. The complete results with std. can be viewed in the Appendix. Moreover, with the proposed rehearsal selection strategy (Ours+RehSel), we have our $A_\infty$ improved, which means the selected example according to their influence has a clear ability for reducing catastrophic forgetting. With our Rehearsal Selection (RehSel) strategy, we have an improvement of 0.77 on $A_\infty$, but $A_1$ and $A_m$ have uncertain performance. This means better memory may bring in worse task conflict.

## 6.3 Analysis of Dataset Influence on SP

In Fig. 3, we count the example with positive/negative influences on old task (S), new task (P), and total SP in Split-CIFAR-10. At each task after task 2, we have 500 fixed-size memory and 10,000 new task data. We first find that most data of old tasks has a positive influence on S and a negative influence on P, while most data of new tasks has a positive influence on P and a negative influence on S. Even so, some data in both new and old tasks has the opposite influence. Then, for the total SP influence, most of memory data has positive influence. In contrast, examples of new tasks have near equal number of positive and negative SP influence. Thus, by clustering and storing examples via higher influence to rehearsal memory buffer, the old knowledge can be kept. By dividing all example influences equally into 5 groups from the minimum to the maximum, we find that most examples have mid-level influence, and server as the main part of the dataset. Also, the numbers of examples with large positive and negative influence are small, which means unique examples are few in the dataset. The observations suggest the example difference should be used to improve model training.

## 6.4 Analysis on SP Pareto Optimum

In this paper, we propose to convert the S-aware and P-aware influence fusion into a DOO problem and use the MGDA to guarantee the fused solution is an SP Pareto optimum. As shown in Fig. 4, we show the comparison of the First Accuracy and Final Accuracy coordinate visualization for all compared methods. We also evaluate with only stability-aware (Ours only S) and with only Plasticity-aware (Ours only P) influence. Obviously, with only one kind of influence, our method can already get better SP than other methods. The integration of two kinds of influence yield an even more balanced SP. On the other hand, the existing methods cannot approach the SP Pareto front well.

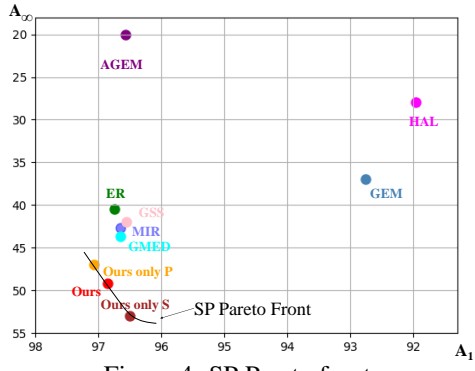

Figure 4: SP Pareto front.

## 6.5 Training Time

We list the training time of one-step update and total update overhead for all compared methods for Split CIFAR-10 dataset. In one-step update, we evaluate all methods

Table 2: Comparison of training time [s] on CIFAR-10.

| Method | ER | GSS | AGEM | HAL | MIR | GMED | GEM | MetaSP |
|---|---|---|---|---|---|---|---|---|
| One-Step | 0.013 | 0.015 | 0.029 | 0.043 | 0.077 | 0.093 | 0.290 | 0.250 |
| Total | 2685 | 2672 | 3812 | 5029 | 7223 | 8565 | 24768 | 5898 |

with a batch on one update. Our method takes more time than other methods except for GEM, because of the pseudo update, backward on perturbation and influence fusion. To guarantee the efficiency, we utilize our proposed method only in the last 5 epochs among the total, and the rest are naive fine-tuning (See details in the Appendix). The results show the strategy is as fast as other light-weight methods but achieve huge improvement on SP. We also use this setting for the comparison in Table 1.

## 7 Conclusion

In this paper, we proposed to explore the example influence on Stability-Plasticity (SP) dilemma in rehearsal-based continual learning. To achieve that, we evaluated the example influence via small perturbation instead of the computationally expensive Hessian-like influence function and proposed a simple yet effective MetaSP algorithm. At each iteration in CL training, MetaSP builds a pseudo update and obtains the S- and P-aware example influence in batch level. Then, the two kinds of influence are combined via an SP Pareto optimal factor and can support the regular model update. Moreover, the example influence can be used to optimize rehearsal selection. The experimental results on three popular CL datasets verified the effectiveness of the proposed method. We list the limitation of the proposed method. (1) The proposed method relies on rehearsal selection, which may affect privacy and extra storage is needed. (2) The proposed method is not fast enough for online continual learning. In most situations, however, we can leverage our training tricks to reduce the time. (3) Our method is limited in the extremely small memory size. Large memory size means better remembering and an accurate validation set. The proposed method does not perform well when the memory size is extremely small.

## Acknowledgement

This work is financially supported in part by the National Key Research and Development Program of China under Grant (No. 2019YFC1520904) and the Natural Science Foundation of China (Nos. 62072334, 62276182, 61876220). The authors would like to thank constructive and valuable suggestions for this paper from the experienced reviewers and AE.

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
