# OpenReview forum: "Exploring Example Influence in Continual Learning"
_NeurIPS.cc/2022/Conference — NeurIPS 2022 Accept_

### Official Review · Reviewer_Qv4L · 2022-07-05

**Rating:** 7
**Confidence:** 4
**Soundness:** 3 good
**Presentation:** 3 good
**Contribution:** 3 good

**Summary:**

The authors investigate the stability dilemma through the lens of the training data and the selection for the replay buffer.
They evaluate the influence of data points on stability and plasticity using small perturbation to data instead of computing hessian (for computation sake).
The authors use the stability and plasticity influence to optimize the gradient update and the sample selection.
The algorithm is evaluated on CIFAR10, CIFAR100, and Mini-Imagenet and compared with several state-of-the-art algorithms.

Overall I like the paper and the idea. I think, however, that the authors could realize some improvements to make the paper clearer and better.

**Questions:**

Table 1: "ours" method use random sampling selection? Maybe comparing the sampling selection to another example from the literature would make sense to assess the method better.
(cf. eg "An Investigation of Replay-based Approaches for Continual Learning" Bagus et al)

What does that mean that a sample increases plasticity or stability? Beyond the definition provided in section 3, I miss the high-level idea....

l266 "We divide all example influence equally to 5 groups from the minimum to the maximum." how does this division is realized exactly?

l269 "The two observations suggest we should consider the example difference to improve model training." but you do not do it, right? So what is the point of this figure in the paper?

Fig.3: left is the evaluation of the data stored in the memory, and the left is the new data? I think the legend could be more detailed to make it clear from the beginning.

Tab 1: code color is not defined.

Is the data used for validation in the memory buffer used at some point for training?

Table 2: how many seeds lead to those results?

l.69 "4) By considering the example influence, in our experiments on both task- and class-incremental, better S and more stable P can be observed" I do not understand what this contribution is...


********
Some Comments:
**************

"KKT" is never defined.

I would move the task-incremental results in the appendix (task incremental is not very classical to study when using replay, but this is not problematic) and add std of results in the main body.

It would be better to group figures 3 and 4 or to modify figure 4, such as the legend is not in figure 3.

**Limitations:**

It would be nice to see in the table which results are statistically significant.

The legend of several figures could be improved.

**Strengths And Weaknesses:**

Strengths:
The paper is well written and organized, with a detailed description of the method.
Sample selection is a crucial problem to study to improve replay methods.
the method is interesting and looks well-performing.

Weaknesses:
The definition of the metrics is not very clear. "Finished Accuracy" seems to be the classical "Final Accuracy". "First Accuracy" and its relationship with plasticity is not very clear.
Is it the average best performance realized on each task?
Maybe, "backward transfer" from "Gradient Episodic Memory for Continual Learning" Lopez-Paz et al would be more precise in measuring stability.
And "Final performance" would measure the mixture between S and P. I am not sure about what "Mean Average Accuracy" brings here.

---

> ### Author Response · Authors · 2022-08-02
> **Response to Reviewer Qv4L (4)**
>
> ### Comments
>
> > "KKT" is never defined.
>
> **Response:** We have revised the manuscript to highlihgt the "KKT conditions" is the abbreviation for "Karush–Kuhn–Tucker conditions". We have described more details about the KKT conditions of MGDA [1] in our Appendix.
>
> *Refer to MGDA*:
>
> We first introduce the Steepest Gradient Method (SGM) [2] in dual-objective optimization. Given two tasks 1 and 2, the objective of SGM
> is
>
> $$
> d^*, \alpha^*=\arg\min\_{d,\alpha}\quad\alpha+\frac{1}{2} ||d||^2,\quad \text{s.t.}\quad g\_1^\top d\le \alpha, g\_2^\top d\le \alpha,
> $$
>
> where $g\_1$ and $g\_2$ are the gradients for task 1 and 2 specifically.
> The two constraints can be seen as the difference between task gradients and the optimal gradients.
>
> Considering the Lagrange multipliers $\lambda\_1$ and $\lambda\_2$ for the two constraints, we have the dual problem of the above problem as
> $$
> \lambda_1^*, \lambda_2^*=\arg\max_{\lambda_1,\lambda_2}\quad -||\lambda_1 g_1+\lambda_2 g_2||^2,\quad\text{s.t.}\quad \lambda_1+\lambda_2=1,\lambda_1\ge0,\lambda_2\ge 0,
> $$
>
> This is the objective of Eq.(10) of the paper.
> In SGM, the KKT conditions can be writen as
>
> $\lambda\_1^*(g\_1^\top d^*- \alpha^*)=0$
>
> $\lambda\_2^*(g\_2^\top d^*- \alpha^*)=0$
>
> $\lambda\_1^* \ge 0, \lambda\_2^* \ge 0$
>
> $\lambda\_1^* + \lambda\_2^* = 1$
>
> $\lambda\_1^* g\_1+\lambda\_2^* g\_2=d^*$
>
>
> The solution of the dual problem is
> $$
> \lambda_1^*=1-\lambda_2^*=\min\left(\max\left(
> \frac{(g\_2-g\_1)^\top g\_2}{\|g\_2-g\_1\|\_2^2},0\right),1\right)
> $$
> This is the objective of Eq.(11) of the paper.
>
> - [1] Ozan Sener and Vladlen Koltun. Multi-task learning as multi-objective optimization. In NeurIPS, 2018.
> - [2] Jörg Fliege and Benar Fux Svaiter. Steepest descent methods for multicriteria optimization. Mathematical methods of operations research, 2000.
>
> > I would move the task-incremental results in the appendix (task incremental is not very classical to study when using replay, but this is not problematic) and add std of results in the main body.
>
> **Response:** Thank you for your suggestion.
>
> > It would be better to group figures 3 and 4 or to modify figure 4, such as the legend is not in figure 3.
>
> **Response:** We have revised the legend of Fig.3 and Fig.4 in the revised manuscript.
>
>
> ## Response to Limitations
>
> > It would be nice to see in the table which results are statistically significant.
>
> **Response:** To show the statistically sigificant of our experiments, we use T-test (Student's t-test) [1] to evaluate that if the means of two methods' results are significantly different from each other. The detail can be seen in Table 1 of the revised manuscript.
>
> [1] Semenick D. Tests and measurements: The T-test[J]. Strength & Conditioning Journal, 1990, 12(1): 36-37.
>
>
>
> > The legend of several figures could be improved.
>
> **Response:** We have revised the legend of Fig.3 and Fig.4 in the revised manuscript.

---

> > ### Comment · Reviewer_Qv4L · 2022-08-08
> > **Final Answer**
> >
> > I think the authors provided a good answer to my questions and comments and I will increase my rating to accept.
> >
> > Some comments:
> >
> > - I would advise to replace "Finished Accuracy" by "Final Accuracy" since it is supposed to be the same thing and "Final Accuracy" is mostly used in the literature.
> >
> > > "the validation set is randomly sampled from memory buffer and all examples in memory are used for training. "
> >
> > using a validation set for training could lead to overfitting, it should be only for model / HPs selection.
> >
> > > "this is an verification contribution."
> >
> > write it then! :)

---

> > > ### Author Response · Authors · 2022-08-09
> > > **Response to the final comments**
> > >
> > > Thank your for your valuable suggestions!
> > >
> > > > I would advise to replace "Finished Accuracy" by "Final Accuracy" since it is supposed to be the same thing and "Final Accuracy" is mostly used in the literature.
> > >
> > > **Response:** Thank you for your suggestion. We have changed the 'finished acc' to 'final acc' in the paper.
> > >
> > > > "the validation set is randomly sampled from memory buffer and all examples in memory are used for training. "
> > > using a validation set for training could lead to overfitting, it should be only for model / HPs selection.
> > >
> > > **Response:** Sorry for the confusion. About the validation set,
> > > 1. Our validation sets are sampled from the training set or memory, and are used to compute the example influences in training set instead of selecting models or hyperparameters.
> > > 2. The validation sampling is dynamic at each step in order to approach the dataset or memory distributions.
> > > 3. We did not split an independent validation set because the memory buffer is too small, where any split will damage the continual learning and lead to more serious catastrophic forgetting.
> > >
> > > > "this is an verification contribution." write it then! :)
> > >
> > > **Response:** Thank you for your suggestion. We have added this to the paper.

---

> ### Author Response · Authors · 2022-08-02
> **Response to Reviewer Qv4L (3)**
>
> ## Answer to Question
>
> > Table 1: "ours" method use random sampling selection? Maybe comparing the sampling selection to another example from the literature would make sense to assess the method better. (cf. eg "An Investigation of Replay-based Approaches for Continual Learning" Bagus et al)
>
> **Answer:** Yes, "ours" method use random sampling selection. Referring to the suggested work [1], we compare the proposed rehearsal selection. The results are shown in the Appendix, and we have cited the missing suggested paper. We also show the comparison as following table. The results show the superviorty of the proposed rehearsal selection method that using example influence.
>
> [1] Bagus B, Gepperth A. An investigation of replay-based approaches for continual learning[C]//IJCNN, 2021.
>
>
> (**CI: Class increment, TI: Task increment**)
> ||CIFAR10-CI-buffer500 |CIFAR10-TI-buffer500|
> |----|:----:|:----:|
> | |$A\_1 / A\_\infty / A\_m$|$A\_1 / A\_\infty / A\_m$ |
> |NSR Random(Ours)         |96.82 / 49.16 / 67.88  |97.30 / 90.91 / 93.38|
> |NSR+ Random        |96.82 / 49.49 / 67.58	|97.20 / 90.94 / 93.22|
> |NSR Mean           |96.60 / 40.63 / 63.90  |97.18 / 89.18 / 92.18|
> |NSR Intensity(lowest)  |96.70 / 26.07 / 52.64  |96.98 / 85.10 / 89.60|
> |NSR Intensity(highest)	|96.39 / 34.79 / 56.39	|96.92 / 85.79 / 87.60|
> |Ours RehSel w/o Influence            |96.82 / 47.55 / 67.49	|97.15 / 90.33 / 93.07|
> |Ours RehSel w/o Cluster              |96.60 / 38.50 / 62.38	|97.27 / 85.55 / 91.22|
> |Ours RehSel        |96.81 / 50.10 / 68.28  |97.30 / 91.41 / 93.28|
>
> > What does that mean that a sample increases plasticity or stability? Beyond the definition provided in sec. 3, I miss the high-level idea....
>
> **Answer:** In this paper, we study the example influence in continual learning. The two significant ability of continual learning is the Plasticity and Stability. Inspired by previous work [1], we propose to explore the example influence on Plasticity and Stability. With the proposed MetaSP, we can ontain the example influence on S and P. Strengthening the training of examples with both positive S and P,  SP can be improved.
>
> [1] Pang Wei Koh and Percy Liang. Understanding black-box predictions via influence functions. In ICML, 2017.
>
> > l266 "We divide all example influence equally to 5 groups from the minimum to the maximum." how does this division is realized exactly?
>
> **Answer:** We first obtain the maximum (MAX) and minimum (MIN) of all example influences, and divide the value range into $\{[\text{MIN}, \text{MIN} + \cfrac{1}{5}(\text{MAX}-\text{MIN})], [\text{MIN}+\cfrac{1}{5}(\text{MAX}-\text{MIN}), \text{MIN} + \cfrac{2}{5}(\text{MAX}-\text{MIN})],\cdots,[\text{MIN}+\cfrac{4}{5}(\text{MAX}-\text{MIN}), \text{MAX}]\}$. We count the number of examples grounding each range and plot Fig.4.
>
> > l269 "The two observations suggest we should consider the example difference to improve model training." but you do not do it, right? So what is the point of this figure in the paper?
>
> **Answer:** We do use the example difference to improve the CL training.
> - This can be seen in Sec 5. We use the influence form differnt examples to rehearsal selection and model update. To improve the model training, we use the example influence to weight the training in example-level.
> - We use Fig.4 to show the example influence distributions in the CL process. We think this is an important insight of the example differnce in CL. As shown in Fig.4, example influences are not in a similar degree, we need to consider the influence difference from examples to improve the training as mentioned in Sec 5.
>
> > Fig.3: left is the evaluation of the data stored in the memory, and the left is the new data? I think the legend could be more detailed to make it clear from the beginning.
>
> **Answer:** The left subfig is the statistics on the 500 memory data and the right subfig is the statisctics on the 10000 new task data. We have revised the legend of Fig.3 and Fig.4 in the revised manuscript.
>
> > Tab 1: code color is not defined.
>
> **Answer:** We have defined the code color of Table 1 in the revised manuscript. The blue values means the best of compared methods and the red values means the best of our methods.
>
> > Is the data used for validation in the memory buffer used at some point for training?
>
> **Answer:** Sure, the validataion set is randomly sampled from memory buffer and all examples in memory are used for training.
>
> > Table 2: how many seeds lead to those results?
>
> **Answer:** We use 5 seeds in all our experiments.
>
> > l.69 "4) By considering the example influence, in our experiments on both task- and class-incremental, better S and more stable P can be observed" I do not understand what this contribution is...
>
> **Answer:** In our view, this is an verification contribution. We evaluate our methods on both task- and class-incremental continual learning and demonstrate that the S and P influence of an example can be obtained and used to improve continual learning.

---

> ### Author Response · Authors · 2022-08-02
> **Response to Reviewer Qv4L (2)**
>
> In the revised manuscript, we also add the metric BWT in Table 1 to show the forgetting. The results is also shown in the folowing table. The results of BWT can also show the superivority of the proposed method.
>
> |Method|CIFAR10-CI-buffer300|CIFAR10-CI-buffer500|CIFAR10-TI-buffer300|CIFAR10-TI-buffer300|
> |----|:----:|:----:|:----:|:----:|
> ||$A\_1/A\_\infty/BWT/A\_m$ |$A\_1/A\_\infty/BWT/A\_m$|$A\_1/A\_\infty/BWT/A\_m$|$A\_1/A\_\infty/BWT/A\_m$|
> |GDUMB  |- / 36.92 / - / - | - / 44.27 - / - / -|- / 73.22 / - / -| - / 78.06 / - / -|
> |GEM    |93.90/37.51/-70.48/55.43   |92.76/36.95/-69.76/57.36   |96.62/89.34/-9.10/92.49    |96.73/90.42/-7.89/92.93|
> |AGEM   |96.57/20.02/-95.69/45.57   |96.56/20.01/-95.70/46.52   |96.78/85.52/-14.07/90.16   |96.71/86.45/-12.83/90.90|
> |HAL    |91.30/24.45/-83.57/46.34   |91.96/27.94/-80.02/49.05   |91.41/79.90/-14.39/83.78   |92.03/81.84/-12.73/84.19|
> |MIR    |96.70/38.53/-72.72/56.96   |96.65/42.65/-67.50/59.99   |96.76/88.50/-10.33/90.87   |96.73/90.63/-7.62/91.99|
> |GSS    |96.53/35.89/-75.80/54.33   |96.55/41.96/-68.25/58.16   |96.56/88.05/-10.63/90.60   |96.57/90.38/-7.74/92.19|
> |GMED   |96.65/38.12/-73.17/58.92   |96.65/43.68/-66.22/62.56   |96.73/88.91/-9.77/91.20    |96.72/89.72/-8.75/92.10|
> |ER     |96.73/34.19/-78.18/53.72   |96.74/40.45/-70.36/57.69   |96.93/88.97/-9.95/91.12    |96.79/90.60/-7.75/92.28|
> |Ours   |96.87/42.42/-68.07/63.52   |96.82/49.16/-59.58/67.88   |97.10/89.40/-9.63/92.54    |97.30/90.91/-7.99/93.38|
> |Ours+RehSel|96.85/43.76/-66.37/63.69|96.81/50.10/-58.39/68.28  |97.11/89.91/-9.00/92.66    |97.30/91.41/-7.36/93.28|
>
> |Method|CIFAR100-CI-buffer500|CIFAR100-CI-buffer1000|CIFAR100-TI-buffer500|CIFAR100-TI-buffer1000|
> |----|:----:|:----:|:----:|:----:|
> ||$A\_1/A\_\infty/BWT/A\_m$ |$A\_1/A\_\infty/BWT/A\_m$|$A\_1/A\_\infty/BWT/A\_m$|$A\_1/A\_\infty/BWT/A\_m$|
> |GDUMB  |- / 11.11 / - / -          |/15.75 / - / -|- /36.40 / - / -|- / 43.25 / - / -|
> |GEM    |85.28/15.91/-77.07/29.38   |84.28/22.79/-68.32/34.09   |85.53/68.68/-18.72/68.49   |85.24/73.71/-12.81/72.59|
> |AGEM   |85.97/9.31/-85.18/24.60    |85.66/9.27/-84.88/24.67    |85.97/55.28/-34.10/58.23   |85.66/55.95/-33.02/59.96|
> |HAL    |67.33/8.20/-65.70/22.72    |68.06/10.59/-63.86/24.74   |67.64/44.98/-25.17/50.79   |68.62/50.07/-20.62/54.01|
> |MIR    |87.38/13.49/-82.10/28.88   |87.39/17.56/-77.60/32.48   |87.42/66.18/-23.60/67.43   |87.50/71.20/-18.11/71.42|
> |GSS    |86.03/14.01/-80.03/28.00   |86.31/17.87/-76.04/31.82   |86.10/66.80/-21.45/66.55   |86.44/71.98/-16.07/71.00|
> |GMED   |87.18/14.56/-80.69/33.41   |87.29/18.67/-76.24/38.69   |87.30/68.82/-20.53/72.66   |87.49/73.91/-15.10/76.36|
> |ER     |87.23/13.75/-81.65/28.88   |87.33/17.56/-77.52/32.45   |87.29/66.82/-22.74/67.56   |87.40/71.74/-17.40/71.60|
> |Ours   |88.13/18.96/-76.85/38.62   |87.58/24.78/-69.77/45.20   |88.94/70.03/-21.01/74.07   |88.94/75.32/-15.14/78.09|
> |Ours+RehSel|87.81/19.28/-76.14/39.23|87.55/25.72/-68.70/45.48  |88.58/70.81/-19.75/74.24   |89.03/76.14/-14.33/78.27|
>
> |Method|MiniImagenet-CI-buffer500|MiniImagenet-CI-buffer1000|MiniImagenet-TI-buffer500|MiniImagenet-TI-buffer1000|
> |----|:----:|:----:|:----:|:----:|
> ||$A\_1/A\_\infty/BWT/A\_m$ |$A\_1/A\_\infty/BWT/A\_m$|$A\_1/A\_\infty/BWT/A\_m$|$A\_1/A\_\infty/BWT/A\_m$|
> |GDUMB  |- / 6.22 / - / -           |- / 7.15 / - / -          |- / 16.37 / - / -|- / 17.69 / - / -|
> |AGEM   |50.06/10.69/-49.22/22.29   |50.03/10.69/-49.17/22.28   |50.06/18.34/-39.66/28.05   |50.03/18.78/-39.06/28.12|
> |MIR    |51.44/11.07/-50.46/23.65   |51.25/11.32/-49.92/24.09   |51.47/29.10/-27.96/35.20   |51.31/31.39/-24.90/37.24|
> |GSS    |51.63/11.09/-50.67/23.62   |51.35/11.42/-49.91/24.05   |51.64/28.67/-28.72/35.22   |51.40/31.75/-24.56/37.23|
> |GMED   |51.21/11.03/-50.24/24.47   |50.87/11.73/-48.93/25.50   |51.29/30.47/-26.03/37.64   |51.00/32.85/-22.69/39.66|
> |ER     |51.68/11.00/-50.84/23.71   |51.41/11.35/-50.08/24.08   |51.70/28.97/-28.41/35.30   |51.55/31.59/-24.95/37.36|
> |Ours   |51.76/12.48/-49.10/26.50   |50.91/14.43/-45.59/28.47   |52.44/32.59/-24.82/39.38   |52.27/36.25/-20.03/41.59|
> |Ours+RehSel|51.81/12.74/-48.85/26.43|50.96/14.54/-45.52/28.44  |51.73/34.36/-21.71/40.48   |51.47/37.20/-17.83/42.19|

---

> ### Author Response · Authors · 2022-08-02
> **Response to Reviewer Qv4L (1)**
>
> Dear reviewer, we have revised the manuscript and reuploaded the revised version, where contents in cyan are revised. Please download the revised manuscript and appendix.
>
> ## Response to Weakness
>
> > The definition of the metrics is not very clear. "Finished Accuracy" seems to be the classical "Final Accuracy". "First Accuracy" and its relationship with plasticity is not very clear. Is it the average best performance realized on each task? Maybe, "backward transfer" from "Gradient Episodic Memory for Continual Learning" Lopez-Paz et al would be more precise in measuring stability. And "Final performance" would measure the mixture between S and P. I am not sure about what "Mean Average Accuracy" brings here.
>
> **Response:** Sorry for the confusion.
>
> Let us review the metrics and the mentioned BWT metrics.
> - Finished Accuracy ($A\_\infty=\frac{1}{T}\sum\_{t}\sum\_{(x,y)\in\mathcal{D}^\text{tst}\_t}\mathbf{1}(y,{\theta}\_T(x))$): The Finished Accuracy is equivalent to the classical Final Accuracy. Most works used this metrics to evaluate all tasks after all tasks have their training finished. Many previous methods use this metric, such as [1-5].
> - First Accuracy ($A\_1=\frac{1}{T}\sum\_{t}\sum\_{(x,y)\in\mathcal{D}^\text{tst}\_t}\mathbf{1}(y,{\theta}\_t(x))$): This metric is used to evaluate the performance of each task when it was first trained done (Evaluate each new task immediately).
> - Mean Average Accuracy ($A\_\text{m}=\frac{1}{T}\sum\_{t}\left(\frac{1}{t}\sum\_{k\le t}\sum\_{(x,y)\in\mathcal{D}^\text{tst}\_k}\mathbf{1}(y,{\theta}\_t(x))\right)$): This metric computes along CL process, indicating the SP performance after each task trained done. Many previous methods use this metric in the form of line figure, such as [6-8].
> - Backward Transfer (BWT from GEM): ($BWT=\frac{1}{T-1}\sum^{T-1}\_{t=1}\sum\_{(x,y)\in\mathcal{D}^\text{tst}\_t}\left(\mathbf{1}(y,{\theta}\_T(x))-\mathbf{1}(y,{\theta}\_t(x))\right)=\frac{T}{T-1}(A\_\infty-A\_1)$) This metric is the performance drop from their first accuracy to the final accuracy of each task.
>
> [1] Jin X, Sadhu A, Du J, et al. Gradient-based Editing of Memory Examples for Online Task-free Continual Learning[J]. Advances in Neural Information Processing Systems, 2021, 34: 29193-29205.
>
> [2] Prabhu A, Torr P H S, Dokania P K. Gdumb: A simple approach that questions our progress in continual learning[C]//European conference on computer vision. Springer, Cham, 2020: 524-540.
>
> [3] Buzzega P, Boschini M, Porrello A, et al. Rethinking experience replay: a bag of tricks for continual learning[C]//2020 25th International Conference on Pattern Recognition (ICPR). IEEE, 2021: 2180-2187.
>
> [4] Lopez-Paz D, Ranzato M A. Gradient episodic memory for continual learning[J]. Advances in neural information processing systems, 2017, 30.
>
> [5] Chaudhry A, Ranzato M A, Rohrbach M, et al. Efficient Lifelong Learning with A-GEM[C]//International Conference on Learning Representations. 2018.
>
> [6] van de Ven G M, Siegelmann H T, Tolias A S. Brain-inspired replay for continual learning with artificial neural networks[J]. Nature communications, 2020, 11(1): 1-14.
>
> [7] Dhar P, Singh R V, Peng K C, et al. Learning without memorizing[C]//Proceedings of the IEEE/CVF conference on computer vision and pattern recognition. 2019: 5138-5146.
>
> [8] Rahaf Aljundi, Lucas Caccia, Eugene Belilovsky, Massimo Caccia, Min Lin, Laurent Charlin, Tinne Tuytelaars[C]. NeurIPS. 2019.
>
> We give an example to show the necessarity of the metric $A\_m$.
>
> |Table 1|After task 1|After task 2| After task 3|
> |--|--|--|--|
> |Task 1|90|85|70|
> |Task 2||85|80|
> |Task 2|||90|
>
> |Table 2|After task 1|After task 2| After task 3|
> |--|--|--|--|
> |Task 1|90|80|70|
> |Task 2||85|80|
> |Task 2|||90|
>
> With the definition, we have
> - Table 1: $A\_1=88.3$, $A\_\infty=80$, $A\_m=83.3$, $BWT=-12.5$
> - Table 2: $A\_1=88.3$, $A\_\infty=80$, $A\_m=82.5$, $BWT=-12.5$
>
> In the two tables, we have same $A\_1$, $A\_\infty$ and BWT but different $A\_m$. Obviously, Table 1 outperforms Table 2. Because an CL model will be deployed at any time, for example right after task 2 instead task 3. $A\_m$ fits this situation because this metric considers all starting, end value of all tasks at any step. As a result, to evaluate the comprehensive ability on both S and P, we suggest $A\_m$.

---

### Official Review · Reviewer_MauF · 2022-07-08

**Rating:** 7
**Confidence:** 4
**Soundness:** 3 good
**Presentation:** 2 fair
**Contribution:** 3 good

**Summary:**

The paper investigates the question of how important each example is for plasticity and stability in continual learning. The authors motivate their approach from the perspective of the Influence Function which was previously used as an explainability method. However, since computing the actual Influence Function requires computing an inverse of the Hessian, the authors decide to instead propose MetaSP, a two-level optimization technique, which aims to approximate the Influence Function. Equipped with this, one can find a loss balancing factor $\gamma$ which is Pareto optimal with respect to stability and plasticity, and then use this balanced loss for training as well as choosing which examples to keep in the buffer. Finally, the authors show an empirical evaluation on multiple datasets and perform additional analysis of the method.

**Questions:**

I would like to ask the authors about the issues mentioned in the "weaknesses" section above. In particular, I'd like to highlight he first one:
do you have theoretical or empirical proof of how well your approach approximates the results obtained from the Importance Function?

**Limitations:**

I don't think a discussion on potential negative societal impact is needed.
The authors do not list the limitations of their method and in my opinion, a section like that should be added. Just to name a few:
- The method relies on experience replay, so it cannot be applied in settings where data privacy is crucial.
- The proposed approach is rather slow (2x slower than ER even with the "use MetaSP only in the last 5-epoch" trick), so it might be an issue in some settings.
- The method was implemented only for classification tasks. How hard would it be to extend it to other problems?

**Strengths And Weaknesses:**

Strengths:
- The idea of disentangling plasticity and stability through IF-like approach and then using this information to get better performance seems novel and intriguing.
- The empirical evaluation is quite thorough and the proposed method performs well. In particular, I appreciate including the training time as the algorithm seems to be computationally complex.
- The ablation studies are useful and give a deeper insight into the method. It's interesting to see how the number of examples with a positive impact on plasticity and stability changes with time.

Weaknesses:
- The major weakness of the paper from my perspective is that in the end, I'm not sure how well the Importance Function motivation connects with the final MetaSP algorithm. Intuitively, these two seem connected, but a theoretical or empirical argument confirming that would be very useful. In particular, do you know how well the MetaSP approach approximates the gradient from Equation (6)? Can we show their equivalence, either theoretically or by running some experiments on toy problems? The authors hint at this kind of result in Section 4.1, but no hard proof is given.
- The fact that the algorithm is actually applied only on the last 5 epochs out of 50 should be highlighted. This is only mentioned in a single subsection near the end of the paper but seems quite crucial. Have you checked how well the other methods perform in this scenario? For example, what happens if we use fine-tuning for 45 epochs and then ER for 5 epochs?
- The presentation could be improved, as the paper is hard to read at times. There are some minor mistakes (see the Minor comments section below) and in general I think the paper could be more self-contained. In particular, checking the previous works was necessary for me to understand Section 5.1 (SP Pareto Optimal), as the authors do not list the KKT conditions and do not really explain what the result of Eq (10) is. Grammar could also be improved, but for the most part, the mistakes do not make it harder to understand the paper so this is not an important issue.
- I'm a bit skeptical whether K-Means on the input space learns anything more than just basic background colors. Have you checked how the examples get clustered? What happens if you do not perform clustering at all (i.e. 1 cluster) or just cluster the examples randomly?
- The paper does not list limitations although in the checklist the authors claim that it does.


Minor comments:
Line 22: "robust CL system should achieve outstanding S while keeping a stable P" - that statement might be a bit controversial as it depends on your desiderata - you might want a model that is plastic but can remember the past moderately well.
Line 96: What do you mean by performance? You are using the $p$ symbol which suggests probability, but that's a bit confusing.
Line 120: "Hessian [...] assumed as positive definite" - that's a pretty strong assumption if we're not close to the minimum (i.e. during the training).
Line 121: "Always out-of-memory" - that's a bit informal, given enough storage or a small number of parameters I think this operation should be tractable. Of course, I agree that in most cases this statement is true, but I would suggest using more precise wording.
Line 168: might be useful to state those KKT conditions
Line 225: I would suggest, when possible, reusing metrics from previous CL papers, as introducing new metrics make it difficult to compare between papers.
Figure 3: At what point is each of these plots generated? I.e. these metrics are dependent on the parameters $\theta$, so which $\theta$ are we using? The one from the end of the current task?
Figure 4: I don't quite understand this figure. Why are there three bars per task and why there are two separate plots?
Figure 5: How was the Pareto front plotted?
Table 2: I assume you mean "training time" rather than "implementation time"?


**EDIT:** Authors' responses and improvements made to the paper convinced me to increase my score to Accept.

---

> ### Author Response · Authors · 2022-08-02
> **Response to Reviewer MauF (4)**
>
> ## Answer to Question
>
> > I would like to ask the authors about the issues mentioned in the "weaknesses" section above. In particular, I'd like to highlight he first one: do you have theoretical or empirical proof of how well your approach approximates the results obtained from the Importance Function?
>
> **Answer:** See the responese in weakness.
>
> ---
>
> ## Response to Limitations
>
> > I don't think a discussion on potential negative societal impact is needed. The authors do not list the limitations of their method and in my opinion, a section like that should be added.
> > - The method relies on experience replay, so it cannot be applied in settings where data privacy is crucial.
> > - The proposed approach is rather slow (2x slower than ER even with the "use MetaSP only in the last 5-epoch" trick), so it might be an issue in some settings.
> > - The method was implemented only for classification tasks. How hard would it be to extend it to other problems?
>
> **Reponse:** We have discussed the limitation of the method in the above weakness.
>
> - Beyond classification task:
> Our method is not related to the network structure and output head or design specific loss function. Our methods can be implemented to any other tasks such as semantic segmentation, detection... by modifying these parts.

---

> ### Author Response · Authors · 2022-08-02
> **Response to Reviewer MauF (3)**
>
> ### Minor comments:
>
> > Line 22: "robust CL system should achieve outstanding S while keeping a stable P" - that statement might be a bit controversial as it depends on your desiderata - you might want a model that is plastic but can remember the past moderately well.
>
> **Reponse:** Yes, it is. We have revised the confused representation.
>
> > Line 96: What do you mean by performance? You are using the $p$ symbol which suggests probability, but that's a bit confusing.
>
> **Reponse:** $p$ means the performance (accuracy in classification), this can be seen in Definition 1 of paper.
>
> > Line 120: "Hessian [...] assumed as positive definite" - that's a pretty strong assumption if we're not close to the minimum (i.e. during the training).
>
> **Reponse:** Sorry for the confusion, we recheck the IF [1] and remove the assumption. We miswrote the assumption in [1] where the problem is convex. In general, in a non-convex model, the Hessian may be not positive definite.
>
> [1] Pang Wei Koh and Percy Liang. Understanding black-box predictions via influence functions. In ICML, 2017.
>
>
> > Line 121: "Always out-of-memory" - that's a bit informal, given enough storage or a small number of parameters I think this operation should be tractable. Of course, I agree that in most cases this statement is true, but I would suggest using more precise wording.
>
> **Reponse:** Sorry for the confusion. We have revised the confused representation to "maybe out-of-memory".
>
> > Line 168: might be useful to state those KKT conditions
>
> **Reponse:** Thank you. The detail can be seen in the response above.
>
> > Line 225: I would suggest, when possible, reusing metrics from previous CL papers, as introducing new metrics make it difficult to compare between papers.
>
> **Reponse:** We have reused previous metrics Backward Transfer (evaluate the performance drop, i.e., forgetting), Finished Accuracy and Mean Average Accuracy in Table 1 of the revised manuscript.
>
> > Figure 3: At what point is each of these plots generated? I.e. these metrics are dependent on the parameters $θ$, so which $θ$ are we using? The one from the end of the current task?
>
> **Reponse:** We plot the figure using the updating $θ$, i.e., the $θ$ is changing. Each example got multiple influence in different mini-batchs. And the final influence is the average of all corresponding batchs and epochs.
>
>
> > Figure 4: I don't quite understand this figure. Why are there three bars per task and why there are two separate plots?
>
> **Reponse:** Sorry for the confusion. We have revised the confusion in the manuscript.
> - Left plot: example influence of old task (memory buffer 500); Right plot: example influence of new task (10000 for Split-CIFAR-10)
> - Three bars: influence to stability, influence to plasticity and influence to both stability and plasticity.
>
> > Figure 5: How was the Pareto front plotted?
>
> **Reponse:** The front is plotted by the fitting of 15 points (5 differnt seeds * 3 ablation types).
>
> > Table 2: I assume you mean "training time" rather than "implementation time"?
>
> **Reponse:** We have revised the confused representation to "training time".

---

> ### Author Response · Authors · 2022-08-02
> **Response to Reviewer MauF (2)**
>
> > The presentation could be improved, as the paper is hard to read at times. There are some minor mistakes (see the Minor comments section below) and in general I think the paper could be more self-contained. In particular, checking the previous works was necessary for me to understand Section 5.1 (SP Pareto Optimal), as the authors do not list the KKT conditions and do not really explain what the result of Eq (10) is. Grammar could also be improved, but for the most part, the mistakes do not make it harder to understand the paper so this is not an important issue.
>
>
> **Reponse:** Sorry for the confusion.
> - We have revised the miner mistakes in our revised manuscript. The details can be seen the response belows.
> - We have described more about the KKT conditions of MGDA [1] in our Appendix.
>
> *Refer to MGDA*:
>
> We first introduce the Steepest Gradient Method (SGM) [2] in dual-objective optimization. Given two tasks 1 and 2, the objective of SGM
> is
> $$
> d^*, \alpha^*=\arg\min\_{d,\alpha}\quad\alpha+\frac{1}{2} ||d||^2,\quad \text{s.t.}\quad g\_1^\top d\le \alpha, g\_2^\top d\le \alpha,
> $$
> where $g\_1$ and $g\_2$ are the gradients for task 1 and 2 specifically.
> The two constraints can be seen as the difference between task gradients and the optimal gradients.
>
> Considering the Lagrange multipliers $\lambda\_1$ and $\lambda\_2$ for the two constraints, we have the dual problem of the above problem as
> $$
> \lambda\_1^*, \lambda\_2^*=-\max\_{\lambda\_1,\lambda\_2}||\lambda\_1 g\_1+\lambda\_2 g\_2||^2,\quad\text{s.t.}\quad \lambda\_1+\lambda\_2=1,\lambda\_1\ge0,\lambda\_2\ge 0,
> $$
> This is the objective of Eq.(10) of the paper.
> In SGM, the KKT conditions can be writen as
>
> $$\lambda\_1^*(g\_1^\top d^*- \alpha^*)=0$$
> $$\lambda\_2^*(g\_2^\top d^*- \alpha^*)=0$$
> $$\lambda\_1^* \ge 0, \lambda\_2^* \ge 0$$
> $$\lambda\_1^* + \lambda\_2^* = 1$$
> $$\lambda\_1^* g\_1+\lambda\_2^* g\_2=d^*$$
>
> The solution of the dual problem is
> $$
> \lambda\_1^*=1-\lambda\_2^*=\min\left(\max\left(
> \frac{(g\_2-g\_1)^\top g\_2}{\|g\_2-g\_1\|\_2^2},0\right),1\right)
> $$
> This is the objective of Eq.(11) of the paper.
>
> - [1] Ozan Sener and Vladlen Koltun. Multi-task learning as multi-objective optimization. In NeurIPS, 2018.
> - [2] Jörg Fliege and Benar Fux Svaiter. Steepest descent methods for multicriteria optimization. Mathematical methods of operations research, 2000.
>
> > I'm a bit skeptical whether K-Means on the input space learns anything more than just basic background colors. Have you checked how the examples get clustered? What happens if you do not perform clustering at all (i.e. 1 cluster) or just cluster the examples randomly?
>
> **Reponse:**
>
> - Our K-Means is based on the color. Images with similar color appearance are more like to be clustered.
> - In the Appendix (Table 5), we have evaluated if we do not perform clustering (1 cluster, named RehSel w/o Cluster) or random selection in rehearsal (named Random). The proposed method with both clustering and influence obtains superior performance.
>
> > The paper does not list limitations although in the checklist the authors claim that it does.
>
> **Reponse:** In the revised paper, we discussed the limitation of the method.
>
> - The proposed method relies on the rehearsal selection. Like most previous rehearsal, storing the raw data may afftect the privacy to some extent and extra storage is needed.
> - The proposed method is not fast enough in online continual learning (only one epoch). In most situation, however, we can leverage our training tricks (finetune + our method) to reduce the time.
> - Our method is limited in the extreme small memory size. Large memory size means better remembering and accurate validation set (sampled from training set). The proposed method does not perform well when the memory size is extreme small. To prove this, we give an extra experiment as follows. When the buffer size is set to 50, our method outperforms ER. However when the buffer size is set to 20, our method has worse performance than ER.
>
>
>
> (**CI: Class increment, TI: Task increment**)
> |Method|CIFAR10-CI-buffer20|CIFAR10-CI-buffer50|CIFAR10-TI-buffer20|CIFAR10-TI-buffer50|
> |----|:----:|:----:|:----:|:----:|
> ||$A\_1/A\_\infty/A\_m$ |$A\_1/A\_\infty/A\_m$|$A\_1/A\_\infty/A\_m$|$A\_1/A\_\infty/A\_m$|
> |ER|97.02/21.87/45.93|97.02/23.37/48.64|97.02/78.60/85.55|97.03/82.63/87.11|
> |Ours|96.80/20.56/45.96|97.24/25.14/50.26|96.99/79.31/86.02|97.32/84.40/89.69|
>
>
> |Method|CIFAR100-CI-buffer20|CIFAR100-CI-buffer50|CIFAR100-TI-buffer20|CIFAR100-TI-buffer50|
> |----|:----:|:----:|:----:|:----:|
> ||$A\_1/A\_\infty/A\_m$ |$A\_1/A\_\infty/A\_m$|$A\_1/A\_\infty/A\_m$|$A\_1/A\_\infty/A\_m$|
> |ER|87.87/9.38/25.76|87.99/9.55/25.91|87.87/42.86/52.99|87.99/48.36/57.84|
> |Ours|89.17/9.46/25.90|89.19/10.33/27.34|89.19/42.83/53.07|89.24/49.19/58.62|

---

> ### Author Response · Authors · 2022-08-02
> **Response to Reviewer MauF (1)**
>
> Dear reviewer, we have revised the manuscript and reuploaded the revised version, where contents in cyan are revised. Please download the revised manuscript and appendix.
>
> ## Response to Weakness
>
> > The major weakness of the paper from my perspective is that in the end, I'm not sure how well the Importance Function motivation connects with the final MetaSP algorithm. Intuitively, these two seem connected, but a theoretical or empirical argument confirming that would be very useful. In particular, do you know how well the MetaSP approach approximates the gradient from Equation (6)? Can we show their equivalence, either theoretically or by running some experiments on toy problems? The authors hint at this kind of result in Section 4.1, but no hard proof is given.
>
> **Reponse:** Sorry for the confusion.
>
> - Eq.(6) shows the chain rule in influence function, that the derivation on the small perturbation can be computed by the product of test gradient, Hessian inverse and the training gradient. The detail can be seen in the Appendix A.2.
> - To evaluate the example influence approximation from our meta method to Hessian Influence Function (Eq. (6)), we build a toy experiment. We use 1000 FMNIST training data and 500 test data. We design a simple fc network with a single hidden layer. The results is the influence from the 1000 training data to 500 test data. We have the following observation: (1) The influence difference of two ways are small; (2) Most examples (965/1000) have the true influence property of compared with Hessian Influence Function.
>
> |Total examples|diff<0.1|diff<0.01|diff<0.001|diff<0.0001|
> |----|----|----|----|----|
> |1000|1000|1000|1000|670|
>
> |Total examples|true positive|true negative|false positive|false negative|
> |----|----|----|----|----|
> |1000|650|315|15|20|
>
> > The fact that the algorithm is actually applied only on the last 5 epochs out of 50 should be highlighted. This is only mentioned in a single subsection near the end of the paper but seems quite crucial. Have you checked how well the other methods perform in this scenario? For example, what happens if we use fine-tuning for 45 epochs and then ER for 5 epochs?
>
> **Reponse:** Thank you for your suggestion. We evaluate the "45 finetune + 5 rehearsal epochs" trick on other rehearsal methods in the following table. The table shows that other rehearsal methods can hardly benifit from the trick, i.e., the performances may got drops compared to training on full 50 epochs. For example, ER achieves better performance in class-incremental learning but gets worse performance in task-incremental learning. We think the performance increase is maybe the less gradient conflict between old and new tasks after 45 epochs. More results about our trick is shown in the Appendix Fig 3(a).
>
>
> (**CI: Class increment, TI: Task increment**)
> |Method |CIFAR10-TI-50epoch  |CIFAR10-TI-45+5epoch|CIFAR10-CI-50epoch |CIFAR10-CI-45+5epoch|
> |----|:----:|:----:|:----:|:----:|
> ||$A\_1/A\_\infty/A\_m$ |$A\_1/A\_\infty/A\_m$|$A\_1/A\_\infty/A\_m$ |$A\_1/A\_\infty/A\_m$|
> |GEM    |96.62/89.34/92.49  |96.64/89.04/91.41	|93.90/37.51/55.43|95.18/35.91/52.54|
> |AGEM   |96.78/85.52/90.16|96.62/66.83/79.11	|96.57/20.02/45.57|96.62/19.65/44.10|
> |MIR    |96.76/88.50/90.87|96.87/86.04/88.95	|96.70/38.53/56.96|96.03/36.90/55.67|
> |GSS    |96.56/88.05/90.60|96.96/86.42/89.13    |96.53/35.89/54.33|96.75/36.78/54.32|
> |GMED   |96.73/88.91/91.20|96.72/89.20/91.66	    |96.65/38.12/58.92|96.73/38.12/58.89|
> |ER     |96.93/88.97/91.12|96.92/85.46/88.78	|96.73/34.19/53.72|96.82/36.95/55.07|
> |Ours   ||97.10/89.40/92.54  ||96.87/42.42/63.52|
> |Ours+RehSel||97.11/89.91/92.66|  |96.85/43.76/63.69|

---

> > ### Comment · Reviewer_MauF · 2022-08-05
> > **Response to the authors**
> >
> > Thank you for the thorough response and modifications made to the paper. Overall, I'm satisfied by the response and I decided to increase my score to "Accept".
> >
> > I don't have any comments for most of the points made by the authors in the reply except for the Hessian approximation experiment (the first response). I appreciate this experiment -- was it added to the paper as well? Also, one more comment -- the diff metrics you show by themselves are not easily interpretable as it is hard to say whether a difference of magnitude 1e-4 is small or big.  I would recommend focusing on the second table (True Positives, etc) or adding some kind of baseline so that the numbers are easier to compare.
> >
> > Another minor comment, Figure 3 (bottom) was definitely improved in the paper, but I still don't get the colorbar/equal proportion part. Could you explain in a little more detail, e.g. in the Figure caption?

---

> > > ### Author Response · Authors · 2022-08-07
> > > **Response to Reviewer MauF**
> > >
> > > > I appreciate this experiment -- was it added to the paper as well? Also, one more comment -- the diff metrics you show by themselves are not easily interpretable as it is hard to say whether a difference of magnitude 1e-4 is small or big. I would recommend focusing on the second table (True Positives, etc) or adding some kind of baseline so that the numbers are easier to compare.
> > >
> > > **Response:** Thanks for your suggestion. We follow the comments to focus on the second table. In detail, we evaluate on the above toy experiments (1000 training data influence 500 test data) with three extra baselines. The three baselines use different ways to approximate inverse Hessian, and the results are shown in the following table and the Appendix. The results show the proposed method has better approximation rate compared with other inverse Hessian approximation methods.
> > >
> > >
> > >
> > > |Method|Total examples|true positive|true negative|false positive|false negative|
> > > |----|----|----|----|----|----|
> > > |Larsen [1]|1000|552|304|26|118|
> > > |Luketina [2]|1000|533|313|17|137|
> > > |Neumann Series [3]|1000|642|282|48|28|
> > > |Ours|1000|650|315|15|20|
> > >
> > > [1] Larsen J, Hansen L K, Svarer C, et al. Design and regularization of neural networks: the optimal use of a validation set[C]//Neural Networks for Signal Processing VI. Proceedings of the 1996 IEEE Signal Processing Society Workshop. IEEE, 1996: 62-71.
> > >
> > > [2] Luketina J, Berglund M, Greff K, et al. Scalable gradient-based tuning of continuous regularization hyperparameters[C]//ICML, 2016
> > >
> > > [3] Lorraine J, Vicol P, Duvenaud D. Optimizing millions of hyperparameters by implicit differentiation[C]//AISTATS. 2020.
> > >
> > > > Another minor comment, Figure 3 (bottom) was definitely improved in the paper, but I still don't get the colorbar/equal proportion part. Could you explain in a little more detail, e.g. in the Figure caption?
> > >
> > > **Response:** Sorry for the confusion, we have revised Figure 3 and the caption to clarify the confused colorbar and proportion.

---

### Official Review · Reviewer_Mnn3 · 2022-07-11

**Rating:** 6
**Confidence:** 3
**Soundness:** 3 good
**Presentation:** 2 fair
**Contribution:** 3 good

**Summary:**

The method augments rehearsal-based methods for continual learning. At the heart of the method is a measurement of the influence of each example on the stability and the plasticity of the algorithm. First, the authors introduce a method for estimating said influence. Second, they present a method for using the influence to regularise the parameter updates. Third, they show how to use the influence in order to select which examples of past tasks to store in memory.

**Questions:**

Do you anticipate that your method would work on sequences of tasks with different input domains? For instance, if you had FMNIST classification mixed with CIFAR10.

**Limitations:**

I did not see a discussion on the limitations of the method. I’d be curious to better understand the number of tasks the method can handle before it breaks, possibly as a function of the memory size.

**Strengths And Weaknesses:**

The paper introduced an interesting direction of combining research on Example Influence with Continual Learning. It appears that the methods they introduce are novel and well motivated. Finally, the experiments show good improvement over the many rehearsal-based baselines.

However, I feel that the paper’s presentation can benefit from another iteration. There are many sentences which need to be improved and that at the moment hinder the readability of the paper. E.g. lines 44, 52, 190, and 203. Moreover, I was not left with the impression that the presentation of the background material, in particular on influence functions, was satisfactory. For instance, I am confused by eq. (6) and the use of iff.
Apart from the presentation, it’d be good if the experiments also introduced baselines which represent other CL directions, s.a. regularisation-based methods.

---

> ### Author Response · Authors · 2022-08-02
> **Response to Reviewer Mnn3 (2)**
>
> ## Answer to Question
>
> > Do you anticipate that your method would work on sequences of tasks with different input domains? For instance, if you had FMNIST classification mixed with CIFAR10.
>
> **Answer:** We evaluate the results of mixed FMNIST and CIFAR-10, and treat them as two tasks (10 + 10 classes for class increment and 1 + 1 tasks for task increment). The comparisons with SOTA can be seen in the following Table, and the results show the superiority of the proposed on the mixed-domain.
>
> (**CI: Class increment, TI: Task increment**)
> |Method|FMNIST-CIFAR10-CI-buffer100|FMINST-CIFAR10-TI-buffer100|
> |----|:----:|:----:|
> ||$A\_1/A\_\infty/A\_m$ |$A\_1/A\_\infty/A\_m$|
> |gdumb	    |-/27.74/-	            |-/29.44/-|
> |agem	    |90.39/67.88/80.98   |90.39/77.51/85.80|
> |hal	    |67.98/25.96/58.90   |67.98/43.58/67.71|
> |mir	    |91.32/64.02/79.05   |91.32/72.57/83.33|
> |gss	    |91.89/61.20/77.70   |91.89/74.78/84.49|
> |gmed	    |91.46/66.15/80.12   |91.46/78.22/86.15|
> |ER         |91.42/60.77/77.43   |91.42/73.17/83.63|
> |Ours       |92.57/75.17/84.63   |92.57/80.72/87.41|
> |Ours+RehSel|92.90/75.85/85.13   |92.90/81.25/87.84|
>
> ---
>
> ## Response to Limitation
>
> > I did not see a discussion on the limitations of the method.
>
> **Reponse:** In the revised paper, we discussed the limitation of the method.
>
> - The proposed method relies on the rehearsal selection. Like most previous rehearsal, storing the raw data may affect the privacy to some extent and extra storage is needed.
> - The proposed method is not fast enough in online continual learning (only one epoch). In most situation, however, we can leverage our training tricks (finetune + our method) to reduce the time.
> - Our method is limited in the extreme small memory size. Large memory size means better remembering and accurate validation set (sampled from training set). The proposed method does not perform well when the memory size is extreme small. To prove this, we give an extra experiment as follows. When the buffer size is set to 50, our method outperforms ER. However when the buffer size is set to 20, our method has worse performance than ER.
>
>
> (**CI: Class increment, TI: Task increment**)
> |Method|CIFAR10-CI-buffer20|CIFAR10-CI-buffer50|CIFAR10-TI-buffer20|CIFAR10-TI-buffer50|
> |----|:----:|:----:|:----:|:----:|
> ||$A\_1/A\_\infty/A\_m$ |$A\_1/A\_\infty/A\_m$|$A\_1/A\_\infty/A\_m$|$A\_1/A\_\infty/A\_m$|
> |ER|97.02/21.87/45.93|97.02/23.37/48.64|97.02/78.60/85.55|97.03/82.63/87.11|
> |Ours|96.80/20.56/45.96|97.24/25.14/50.26|96.99/79.31/86.02|97.32/84.40/89.69|
>
>
> |Method|CIFAR100-CI-buffer20|CIFAR100-CI-buffer50|CIFAR100-TI-buffer20|CIFAR100-TI-buffer50|
> |----|:----:|:----:|:----:|:----:|
> ||$A\_1/A\_\infty/A\_m$ |$A\_1/A\_\infty/A\_m$|$A\_1/A\_\infty/A\_m$|$A\_1/A\_\infty/A\_m$|
> |ER|87.87/9.38/25.76|87.99/9.55/25.91|87.87/42.86/52.99|87.99/48.36/57.84|
> |Ours|89.17/9.46/25.90|89.19/10.33/27.34|89.19/42.83/53.07|89.24/49.19/58.62|
>
>
> > I’d be curious to better understand the number of tasks the method can handle before it breaks, possibly as a function of the memory size.
>
> **Response:** We enlarge the task number of CIFAR-100, i.e., we split it into 50 tasks (2 classes per task). The results are shown in the following table. We observe the performance of our method still outperfoms ER especially in class-incremental CL.
>
> (**CI: Class increment, TI: Task increment**)
> |Method|CIFAR100-CI-buffer500|CIFAR100-CI-buffer1000|CIFAR100-TI-buffer500|CIFAR100-TI-buffer1000|
> |----|:----:|:----:|:----:|:----:|
> ||$A\_1/A\_\infty/A\_m$ |$A\_1/A\_\infty/A\_m$|$A\_1/A\_\infty/A\_m$|$A\_1/A\_\infty/A\_m$|
> |ER|95.66/3.90/10.53|95.58/4.92/12.18|95.20/77.48/78.75|95.72/81.67/81.66|
> |Ours|96.80/20.56/45.96|97.24/25.14/50.26|96.99/79.31/86.02|97.32/84.40/89.69|

---

> ### Author Response · Authors · 2022-08-02
> **Response to Reviewer Mnn3 (1)**
>
> Dear reviewer, we have revised the manuscript and reuploaded the revised version, where contents in cyan are revised. Please download the revised manuscript and appendix.
>
> ## Response to Weakness
>
> > However, I feel that the paper’s presentation can benefit from another iteration. There are many sentences which need to be improved and that at the moment hinder the readability of the paper. E.g. lines 44, 52, 190, and 203.
>
> **Reponse:** Sorry for the confusion, we have revised the mentioned sentences in the revised manuscript.
>
> > Moreover, I was not left with the impression that the presentation of the background material, in particular on influence functions, was satisfactory. For instance, I am confused by eq. (6) and the use of iff. Apart from the presentation, it’d be good if the experiments also introduced baselines which represent other CL directions, s.a. regularisation-based methods.
>
> **Reponse:** Sorry for the confusion.
> - Background on influence function: Referring to [1], influence function is a classic technique from robust statistics to trace a model’s prediction through the learning algorithm and back to its training data, thereby identifying the training data most responsible for a given prediction.
> - Eq.(6): This equation indicates that the example influence $ I(\mathcal{D}^\mathrm{tst},\mathcal{B})$ is reflected in the derivative $\nabla\_{\mathbf{E}}\ell(\mathcal{D}^\mathrm{tst},\hat{{\theta}}\_{\mathbf{E}, x})\big|\_{\mathbf{E}=\mathbf{0}}$. In detail
> $$
> \mathbf{I}(\mathcal{D}^\mathrm{tst},\mathcal{B})= \frac{\partial l(\mathcal{D}^\mathrm{tst},\hat{ {\theta}}\_{\mathbf{E}, \mathcal{B}})}{\partial \mathbf{E}}\bigg|\_{\mathbf{E}=\mathbf{\mathbf{0}}}
> =\frac{\partial l(\mathcal{D}^\mathrm{tst},\hat{{ {\theta}}})}{\partial{ {\theta}}}\cdot\frac{\partial\hat{{ {\theta}}}\_{\mathbf{E}, \mathcal{B}}}{\partial \mathbf{E}}\bigg|\_{\mathbf{E}=\mathbf{\mathbf{0}}}- \nabla\_{{ {\theta}}}l(\mathcal{D}^\mathrm{tst},\hat{{ {\theta}}})\mathbf{H}^{-1}\nabla^\top\_{{ {\theta}}}\mathbf{L}(\mathcal{B},\hat{{ {\theta}}}),
> $$
> where
> $$
> \mathbf{H}=\frac{1}{|\mathcal{B}|}\sum\_{x\_i\in\mathcal{B}}\nabla\_{ {\theta}}^2\ell(x\_i,\hat{ {\theta}}\_{\epsilon\_i}),
> \quad
> \frac{\partial\hat{{ {\theta}}}\_{\mathbf{E}, \mathcal{B}}}{\partial \mathbf{E}}\bigg|\_{\mathbf{E}=\mathbf{0}}=-\mathbf{H}^{-1}\left[  \frac{\partial \mathbf{L}}{\partial {\theta}} \right]^\top.
> $$
> More details can be seen in Appendix A.2.
> - The use of $\iff$ in Eq.(6): We have modified $\iff$ to $\stackrel{\text{def}}{=}$ to represent the meaning of "define". We have added the description in the revised manuscript.
> - We compare other regularization-based methods including EWC, SI, and LwF, the results (average in 5 seeds) are shown in the following table. The results show that regularization-based methods perform not well in class-incremental CL, but have comparable performance to other rehearsal methods in task-incremental CL.
>
> |Method|CIFAR10-ClassIncrement|CIFAR10-TaskIncrement|
> |----|----|----|
> ||$A\_1/A\_\infty/A\_m$ |$A\_1/A\_\infty/A\_m$|
> |EWC|93.89/18.99/43.12|93.89/73.01/76.20|
> |SI|96.41/19.59/44.10|96.53/68.63/77.41|
> |LwF|96.61/19.61/44.03|96.71/74.03/81.20|
> |Ours(buffer 500)|96.82/49.16/67.88|97.31/90.91/93.38|
> |Ours+RehSel(buffer 500)|96.81/50.10/68.28|97.30/91.41/93.28|
>
> |Method|CIFAR100-ClassIncrement|CIFAR100-TaskIncrement|
> |----|----|----|
> ||$A\_1/A\_\infty/A\_m$ |$A\_1/A\_\infty/A\_m$|
> |EWC|80.48/8.43/23.46|80.25/35.96/39.20|
> |SI|82.34/9.20/22.92|82.34/36.96/44.63|
> |LwF|83.23/9.27/22.91|83.23/50.53/56.27|
> |Ours(buffer 500)|88.13/18.96/38.62|88.94/70.03/74.07|
> |Ours+RehSel(buffer 500)|87.81/19.28/39.23|88.58/70.81/74.24|
>
> |Method|Mini-Imagenet-ClassIncrement|Mini-Imagenet-TaskIncrement|
> |----|----|----|
> ||$A\_1/A\_\infty/A\_m$ |$A\_1/A\_\infty/A\_m$|
> |EWC|46.07/10.00/21.41|46.27/21.22/30.01|
> |SI|36.80/6.02/19.15|36.80/27.90/33.54|
> |LwF|52.05/11.13/23.05|52.05/31.74/37.29|
> |Ours(buffer 500)|51.76/12.48/26.50|52.44/32.59/39.38|
> |Ours+RehSel(buffer 500)|51.81/12.74/26.43|51.73/34.36/40.48|
>
> [1] Pang Wei Koh and Percy Liang. Understanding black-box predictions via influence functions. In ICML, 2017.

---

### Meta-Review · Area_Chair_dYgR · 2022-08-21

**Recommendation:** Accept
**Confidence:** Certain

**Metareview:**

There was a consensus among reviewers that this paper should be accepted. The paper investigates an interesting direction of combining research on Example Influence with Continual Learning. The methods they introduce was considered to be novel and well-motivated by the reviewers and the experiments show good improvement over the many rehearsal-based baselines.

**Award:**

No

---

### Decision · Program_Chairs · 2022-09-14

Accept